# A Semantic Generalization of Shannon’s Information Theory and Applications

**DOI:** 10.3390/e27050461

**Published:** 2025-04-24

**Authors:** Chenguang Lu

**Affiliations:** 1Intelligence Engineering and Mathematics Institute, Liaoning Technical University, Fuxin 123000, China; survival99@gmail.com; 2School of Computer Engineering and Applied Mathematics, Changsha University, Changsha 410022, China

**Keywords:** semantic information theory, semantic information measures, information rate–distortion, information rate–fidelity, variational Bayes, free energy principle, maximum information efficiency, portfolio, information value, constraint control

## Abstract

Does semantic communication require a semantic information theory parallel to Shannon’s information theory, or can Shannon’s work be generalized for semantic communication? This paper advocates for the latter and introduces a semantic generalization of Shannon’s information theory (G theory for short). The core idea is to replace the distortion constraint with the semantic constraint, achieved by utilizing a set of truth functions as a semantic channel. These truth functions enable the expressions of semantic distortion, semantic information measures, and semantic information loss. Notably, the maximum semantic information criterion is equivalent to the maximum likelihood criterion and similar to the Regularized Least Squares criterion. This paper shows G theory’s applications to daily and electronic semantic communication, machine learning, constraint control, Bayesian confirmation, portfolio theory, and information value. The improvements in machine learning methods involve multi-label learning and classification, maximum mutual information classification, mixture models, and solving latent variables. Furthermore, insights from statistical physics are discussed: Shannon information is similar to free energy; semantic information to free energy in local equilibrium systems; and information efficiency to the efficiency of free energy in performing work. The paper also proposes refining Friston’s minimum free energy principle into the maximum information efficiency principle. Lastly, it compares G theory with other semantic information theories and discusses its limitation in representing the semantics of complex data.

## 1. Introduction

Although Shannon’s information theory [1] has achieved remarkable success, it faces three significant limitations that restrict its applications to semantic communication and machine learning. First, it cannot measure semantic information. Second, it relies on the distortion function to evaluate communication quality, but the distortion function is subjectively defined and lacks an objective standard. Third, it is challenging to incorporate model parameters into Shannon’s entropy and information formulas. In contrast, machine learning often uses cross-entropy and cross-mutual information involving model parameters. Moreover, the minimum distortion criterion resembles the philosophy of the “absence of fault is a virtue”, whereas a more desirable principle might be “merit outweighing fault is a virtue”. Why does Shannon’s information theory use the distortion criterion instead of the information criterion? This is intriguing.

The study of semantic information gained attention soon after Shannon’s theory emerged. Weaver initiated research on semantic information and information utility [2], and Carnap and Bar-Hillel proposed a semantic information theory [3]. Thirty years ago, the author of this article generalized Shannon’s theory to a semantic information theory [4,5,6]. Now, we call it G theory for short [7]. Earlier contributions to semantic information theories include works by Carnap and Bar-Hillel [3], Dretske [8], Wu [9], and Zhong [10], while contributions after the author’s generalization include those by Floridi [11,12] and others [13]. These theories primarily address natural language information and semantic information measures for daily semantic communication, involving the philosophical foundation of semantic information.

In the past decade, research on semantic communication and semantic information theory has developed rapidly in the following two fields. One field is electronic communication. The demand for the sixth generation high-speed internet has led to the emergence of some new semantic information theories or methods [14,15,16], which pay more attention to electronic semantic communication, especially semantic compression and distortion [17,18,19]. These studies have a high practical value. However, they mainly address electronic semantic information without considering how to measure the information in daily semantic communication, such as information conveyed by a prediction, a GPS pointer, a color perception, or a label. (All the abbreviations with original texts are listed in the back matter).

The other field is machine learning. Now, cross-entropy, posterior cross-entropy *H*(*X*|*Y_θ_*) (roughly speaking, it is Variational Free Energy (VFE) [20,21,22]), semantic similarity, estimating mutual information (MI) [23,24,25], regularization distortion [26], etc., are widely used in the field of machine learning. These information or entropy measures are used to optimize model parameters and latent variables [27,28,29], achieving significant successes. However, machine learning researchers rarely talk about “semantic information theory”. An important reason is that these authors have not found that estimating MI is a special case of semantic MI, and source entropy *H*(*X*) minus VFE equals semantic MI. So, they also did not propose a general measure of semantic information. Machine learning researchers are still unclear about the relationship between estimating MI and Shannon MI. For example, there is controversy over which type of MI needs to be maximized or minimized [23,30,31,32]. Although significant progress has been made in applying deep learning methods to electronic semantic communication and semantic compression [17,18,33], the theoretical explanation is still unclear, resulting in many different methods.

Thirty years ago, the author extended the information rate–distortion function to obtain the information rate–fidelity function *R*(*G*) (where *R* is the minimum Shannon’s MI for the given semantic MI, *G*). The *R*(*G*) has already been applied to image data compression according to visual discrimination [5,7]. Over the past 10 years, the author has incorporated model parameters into the truth function and used it as the learning function [7,32,34], optimizing it with the sample distribution. Machine learning applications include multi-label learning and classification, the maximum MI classification of unseen instances, mixture models [8], Bayesian confirmation [35,36], semantic compression [37], and solving latent variables. Semantic communication urgently requires semantic compression theories like the information rate–distortion theory [38,39,40]. The *R*(*G*) function can intuitively display the relationship between Shannon MI and semantic MI and seems to have universal significance.

Due to the above reasons, the first motivation for writing this article is to combine the above three fields to introduce G theory and its applications, demonstrating that different fields can use the same semantic information measures and optimization methods.

On the other hand, researchers hold two extreme viewpoints on semantic information theory. One view argues that Shannon’s theory suffices, rendering a dedicated semantic information theory unnecessary; at most, semantic distortion needs consideration. This is a common practice in the field of electronic semantic communication. The opposite viewpoint advocates for a parallel semantic information theory alongside Shannon’s framework. Among parallel approaches, some researchers (e.g., Carnap and Bar-Hillel) use only logical probability without statistical probability; others incorporate semantic sources, semantic channels, semantic destinations, and semantic information rate distortion [16].

G theory offers a compromise between these extremes. It fully inherits Shannon’s information theory, including its derived theories. Only the semantic channel composed of truth functions is newly added. Based on Davidson’s truth-conditional semantics [41], truth functions represent the extensions and semantics of concepts or labels. By leveraging the semantic channel, G theory can

Derive the likelihood function from the truth function and source, enabling semantic probability predictions, thereby quantifying semantic information;Replace the distortion constraint in Shannon’s theory with semantic constraints, which include semantic distortion, semantic information quantity, and semantic information loss constraints.

The semantic information measure (i.e., the G measure) does not replace Shannon’s information measure but supplants the distortion metric used to evaluate communication quality. Truth functions can be derived from sample distributions using machine learning techniques with the maximum semantic information criterion, addressing the challenges of defining classic distortion functions and optimizing Shannon channels with an information criterion. A key advantage of generalization over reconstruction is that semantic constraint functions can be treated as new and negative distortion functions, allowing the use of existing coding methods without additional coding considerations for electronic semantic communication.

Because of the above reasons, the second motivation for writing this article is to point out that based on Shannon’s information theory, we only need to replace the distortion constraint with the semantic constraints for semantic communication optimization.

G theory has been continuously improved in recent years, and many conclusions are scattered across about ten articles. Therefore, the author wants to provide a comprehensive introduction so that later generations can avoid detours. This is the third motivation for writing this article.

The existing literature reviews of semantic communication theory mainly introduce the progress of electronic semantic communication [14,15,42,43]. However, this article mainly involves daily semantic communication with its philosophical foundation and machine learning. Although this article mainly introduces G theory, it also compares its differences and similarities with other semantic information theories. Electronic semantic communication theory is different from semantic information theory. The former focuses on one task and involves many theories, whereas the latter focuses on a fundamental theory that involves many tasks. They are complementary, but one cannot replace the other.

The novelty of this article also lies in the introduction of philosophical thoughts behind G theory. In addition to Shannon’s idea, G theory integrates Popper’s views on semantic information, logical probability, and factual testing [44,45], as well as Fisher’s maximum likelihood criterion [46] and Zadeh’s fuzzy set theory [47,48].

The primary purposes of this article are to

Introduce G theory and its applications for exchanging ideas with other researchers in different fields related to semantic communication;Show that G theory can become a fundamental part of future unified semantic information theory.

The main contributions of this article are as follows:It systematically introduces G theory from a new perspective (replacing distortion constraints with semantic constraints) and points out its connections with Shannon information theory and its differences and similarities with other semantic information theories.It systematically introduces the applications of G theory in semantic communication, machine learning, Bayesian confirmation, constraint control, and investment portfolios.It links many concepts and methods in information theory and machine learning for readers to better understand the relationship between semantic information and machine learning.

G theory is also limited, as it is not a complete or perfect semantic information theory. For example, its semantic representation and data compression of complex data cannot keep up with the pace of deep learning.

The remainder of this paper is organized as follows: Section 2 introduces G theory; Section 3 discusses the G measure for electronic semantic communication; Section 4 explores goal-oriented information and information value (in conjunction with portfolio theory); and Section 5 examines G theory’s applications to machine learning. The final section provides discussions and conclusions, including comparing G theory with other semantic information theories, exploring the concept of information, and identifying G theory’s limitations and areas for further research.

## 2. From Shannon’s Information Theory to G Theory

### 2.1. Semantics and Semantic Probabilistic Predictions

Popper pointed out, in his 1932 book, “*The Logic of Scientific Discovery*” [44] (p. 102): the significance of scientific hypotheses lies in their predictive power, and predictions provide information; the smaller the logical probability and the more it can withstand testing, the greater the amount of information it provides. Later, he proposed a logical probability axiom system. He emphasized that a hypothesis has two types of probabilities, statistical and logical probabilities, at the same time [39] (pp. 252–258). However, he had not established a probability system that included both. In Popper’s book, “*Conjectures and Refutations*” [45] (p. 294), he affirmed more clearly that the value of scientific theory lies in the information. We can say that Popper is the earliest researcher of semantic information theory.

The semantics of a word or label encompass both its connotation and extension. Connotation refers to an object’s essential attribute, while extension denotes the range of objects the term refers to. For example, the extension of “adult” includes individuals aged 18 and above, while its connotation is “over 18 years old”. Extensions for some concepts, like “adult”, may be explicitly defined by regulations, whereas others, such as “elderly”, “heavy rain”, “excellent grades”, or “hot weather”, are more subjective and evolve through usage. Connotation and extension are interdependent; we can often infer one from the other.

According to Tarski’s truth theory [49] and Davidson’s truth-conditional semantics [41], a concept’s semantics can be represented by a truth function, which reflects the concept’s extension. For a crispy set, the truth function acts as the characteristic function of the set. For example, if *x* is age, and *y*_1_ is the label of the set {adult}, we denote the truth function as *T*(*y*_1_|*x*), which is also the characteristic function of the set {adult}.

The truth function serves as the tool for semantic probability predictions (illustrated in Figure 1). The formula is(1)$$P(x|y_{1}\;\mathrm{is}\;\mathrm{true})=P(x)T(y_{1}|x)/\sum_{i}P(x_{i})T(y_{1}|x_{i}).$$

If “adult” is changed to “elderly”, the crispy set becomes a fuzzy set, so the truth function is equal to the membership function of the fuzzy set, and the above formula remains unchanged.

The extension of a sentence can be regarded as a fuzzy range in a high-dimensional space. For example, an instance described by a sentence with a subject, object, and predicate structure can be regarded as a point in the Cartesian product of three sets, and the extension of a sentence is a fuzzy subset in the Cartesian product. For example, the subject and the predicate are two people in the same group, and the predicate can be one of “bully”, “help”, etc. The extension of “Tom helps Jone” is an element in the three-dimensional space, and the extension of “Tom helps an old man” is a fuzzy subset in the three-dimensional space. The extension of a weather forecast is a subset in the multidimensional space with time, space, rainfall, temperature, wind speed, etc., as coordinates. The extension of a photo or a compressed photo can be regarded as a fuzzy set, including all things with similar characteristics.

Floridi affirms that all sentences or labels that may be true or false contain semantics and provide semantic information [12]. The author agrees with this view and suggests converting the distortion function and the truth function *T*(*y_j_*|*x*) to each other. To this end, we define*T*(*y_j_*|*x*) ≡ exp[−*d*(*y_j_*|*x*)], *d*(*y_j_*|*x*) ≡ −log*T*(*y_j_*|*x*). (2)
where exp and log are a pair of inverse functions; *d*(*y_j_*|*x*) means the distortion when *y_j_* represents *x_i_*. We use *d*(*y_j_*|*x*) instead of *d*(*x*, *y_j_*) because the distortion may be asymmetrical.

For example, the pointer on a Global Positioning System (GPS) map has relative deviation or distortion; the distortion function can be converted to a truth function or similarity function:*T*(*y_j_*|*x*) = exp[−*d*(*y_j_*|*x*)] = exp[−(*x* − *x_j_*)^2^/(2*σ*^2^)], (3)
where *σ* is the standard deviation; the smaller it is, the higher the precision. Figure 2 shows the mobile phone positioning seen by someone on a train.

According to the semantics of the GPS pointer, we can predict that the actual position is an approximate normal distribution on the high-speed rail, and the red pentagram indicates the maximum possible position. If a person is on a specific highway, the prior probability distribution, *P*(*x*), is different, and the maximum possible position is the place closest to the small circle on the highway.

Clocks, scales, thermometers, and various economic indices are similar to the positioning pointers and can all be regarded as estimates ($y_{j}={\hat{x}}_{j}$) with error ranges or extensions, so they can all be used for semantic probability prediction and provide semantic information. A color perception can also be regarded as an estimate of color or color light. The higher the discrimination of the human eye (expressed by a smaller *σ*), the smaller the extension. A Gaussian function can also be used as a truth function or discrimination function.

### 2.2. The P-T Probability Framework

Carnap and Bar-Hillel only use logical probability. In the above semantic probability prediction, we use the statistical probability, *P*(*x*); the logical probability; and the truth value, *T*(*y*_1_|*x*), which can be regarded as the conditional logical probability. G theory is based on the P-T probability framework, which is composed of Shannon’s probability framework, Kolmogorov’s probability space [50], and Zadeh’s fuzzy sets [47,48].

Next, we introduce the P-T probability framework by its construction process.

**Step 1:** We use Shannon’s probability framework (see the left part of Figure 3). The two random variables, *X* and *Y*, take two values from two domains, ***U*** = {*x*_1_, *x*_2_, …} and ***V*** = {*y*_1_, *y*_2_, …}, respectively. The probability is the limit of the frequency, as defined by Mises [51]. We call it statistical probability. The statistical probability is defined by “=”, such as *P*(*y_j_*|*x_i_*) = *P*(*Y* = *y_j_*|*X* = *x_i_*). Shannon’s probability framework can be represented by a triple (***U***, ***V***, *P*).

**Step 2:** We use domain ***U*** to define the Kolmogorov’s probability of a set (see the right side of Figure 3). Kolmogorov’s probability space can be represented by a triple (***U***, ***B***, *P*), where ***B*** is the Borel field on ***U***, and its element, *θ_j_*, is a subset of ***U***. If we only consider a discrete ***U***, then ***B*** is the power set of ***U***. The probability, *P*, is the probability of a subset of ***U***. It is defined by “∈”, that is, *P*(*θ_j_*) = *P*(*X*∈*θ_j_*). To distinguish it from the probability of elements, we use *T* to denote the probability of a set, so the triple becomes (***U***, ***B***, *T*).

**Step 3:** Let *y_j_* be the label of *θ_j_*, that is, define that there is a bijection between ***B*** and ***V*** (one-to-one correspondence between their elements), and *y_j_* is the image of *θ_j_*. Hence, we obtain the quintuple (***U***, ***V***, *P*, ***B***, *T*).

**Step 4:** We generalize *θ*_1_, *θ*_2_, … to fuzzy sets to obtain the P-T probability framework, which is represented by the same quintuple (***U***, ***V***, *P*, ***B***, *T*).

Because of *y_j_* = “*X* is in *θ_j_*”, *T*(*θ_j_*) equals *P*(*X* = ∈*θ_j_*) = *P*(*y_j_* is true) (according to Tarsiki’s truth theory [49]). So, *T*(*θ_j_*) equals the logical probability of *y_j_*, that is, *T*(*y_j_*) ≡ *T*(*θ_j_*).

Given *X* = *x_i_*, the conditional logical probability of *y_j_* becomes the truth value *T*(*θ_j_*|*x_i_*) of the proposition *y_j_*(*x_i_*), and the truth function *T*(*θ_j_*|*x*) is also the membership function of *θ_j_*. That is,*T*(*y_j_*|*x*) ≡ *T*(*θ_j_*|x) ≡ *m_θ__j_*(*x*). (4)

We also treat *θ_j_* as the model parameter in the parameterized truth function. This makes it easier to establish the connection between the truth function and the likelihood function. According to Davidson’s truth conditional semantics [41], *T*(*θ_j_*|*x*) reflects the semantics of *y_j_*.

The only problem with the P-T probability framework is that the fuzzy set algebra on ***B*** may not follow Boolean algebra operations. The author used fuzzy quasi-Boolean algebra [52] to establish the mathematical model of the color vision mechanism, which can solve this problem to a certain extent. Because, in general, we do not need complex fuzzy set functions *f*(*θ*_1_, *θ*_2_, …), so this problem can be temporarily ignored.

According to the above definition, *y_j_* has two probabilities: *P*(*y_j_*), meaning how often *y_j_* is selected, and *T*(*θ_j_*), meaning how true *y_j_* is. The two are generally not equal. The logical probability of a tautology is one, while its statistical probability is close to zero. We have *P*(*y*_1_) + *P*(*y*_2_) + … + *P*(*y_n_*) = 1, but it is possible that *T*(*y*_1_) + *T*(*y*_2_) + … + *T*(*y_n_*) > 1. For example, the age labels include “adult”, “non-adult”, “child”, “youth”, “elderly”, etc., and the sum of their statistical probabilities is one. In contrast, the sum of their logical probabilities is greater than one because the sum of the logical probabilities of “adult” and “non-adult” alone is equal to one.

According to the above definition, we have(5)$$T(y_{j})\equiv T(\theta_{j})\equiv P(X\in \theta_{j})=\sum_{i}P(x_{i})T(\theta_{j}|x_{i}).$$
This is the probability of a fuzzy event, as defined by Zadeh [48].

We can put *T*(*θ_j_*|*x*) and *P*(*x*) into the Bayes’ formula to obtain the semantic probability prediction formula:(6)$$P(x|\theta_{j})=\frac{T(\theta_{j}|x)P(x)}{T(\theta_{j})},\;T(\theta_{j})=\sum_{i}T(\theta_{j}|x_{i})P(x_{i}).$$

*P*(*x*|*θ_j_*) is the likelihood function *P*(*x*|*y_j_*, *θ*) in the popular method. We use *P*(*x*|*θ_j_*) here because *θ_j_* is bound to *y_j_*. We call the above formula the semantic Bayes’ formula.

Because the maximum value of *T*(*y_j_*|*x*) is 1, from *P*(*x*) and *P*(*x*|*θ_j_*), we derive a new formula:(7)$$T(\theta_{j}|x)=\frac{P(x|\theta_{j})}{P(x)}/\underset{x}{\max[}\frac{P(x|\theta )}{P(x)}].$$

### 2.3. Semantic Channel and Semantic Communication Model

Shannon calls *P*(*X*), *P*(*Y*|*X*), and *P*(*Y*) the source, the channel, and the destination. Just as a set of transition probability functions *P*(*y_j_*|*x*) (*j* = 1, 2, …) constitutes a Shannon channel, a set of truth value functions *T*(*θ_j_*|*x*) (*j* = 1, 2, …) constitutes a semantic channel. The comparison of the two channels is shown in Figure 3. For convenience, we also call *P*(*x*), *P*(*y*|*x*), and *P*(*y*) the source, the channel, and the destination, and we call *T*(*y*|*x*) the semantic channel.

The semantic channel reflects the semantics or extensions of labels, while the Shannon channel indicates the usage of labels. The comparison between the Shannon and the semantic communication models is shown in Figure 4. The distortion constraint is usually not drawn, but it actually exists.

The semantic channel contains information about the distortion function, and the semantic information represents the communication quality, so there is no need to define a distortion function anymore. The purpose of optimizing the model parameters is to make the semantic channel match the Shannon channel, that is, *T*(*θ_j_*|*x*)∝*P*(*y_j_*|*x*) or *P*(*x*|*θ_j_*) = *P*(*x*|*y_j_*) (*j* = 1, 2, …), so that the semantic MI reaches its maximum value and is equal to the Shannon MI. Conversely, when the Shannon channel matches the semantic channel, the information difference reaches the minimum, or the information efficiency reaches the maximum.

### 2.4. Generalizing Shannon’s Information Measure to the Semantic Information G Measure

Shannon MI can be expressed as(8)$$I(X;Y)=\sum_{j}\sum_{i}P(x)P(x|y_{j})\log\frac{P(x_{i}|y_{j})}{P(x_{i})}=H(X)-H(X|Y),$$
where *H*(*X*) is the entropy of *X*, reflecting the minimum average code length. *H*(*X*|*Y*) is the posterior entropy of *X*, reflecting the minimum average code length after predicting *x* based on *y*. Therefore, the Shannon MI means the average code length saved due to the prediction *P*(*x*|*y*).

Replacing *P*(*x_i_*|*y_j_*) on the right side of the log with the likelihood function *P*(*x_i_*|*θ_j_*), we obtain the following semantic MI:(9)$$\begin{array}{l} I(X;Y_{\theta })=\sum_{j}\sum_{i}P(x_{i})P(x_{i}|y_{j})\log\frac{P(x_{i}|\theta_{j})}{P(x_{i})}\\ =\sum_{j}\sum_{i}P(x_{i})P(x_{i}|y_{j})\log\frac{T(\theta_{j}|x_{i})}{T(\theta_{j})}\\ =H(X)-H(X|Y_{\theta })=H(Y_{\theta })-H(Y_{\theta }|X)=H(Y_{\theta })-\bar{d}, \end{array}$$
where *H*(*X*|*Y_θ_*) is the semantic posterior entropy of *x*:(10)$$H(X|Y_{\theta })=-\sum_{j}\sum_{i}P(x_{i},y_{j})\log P(x_{i}|\theta_{j}).$$
Roughly speaking, *H*(*X*|*Y_θ_*) is the free energy, *F*, in the Variational Bayes method (VB) [27,28] and the minimum free energy (MFE) principle [20,21,22]. The smaller it is, the greater the amount of semantic information. *H*(*Y_θ_*|*X*) is called the fuzzy entropy, equal to the average distortion, $\bar{d}$. According to Equation (2), there is(11)$$H(Y_{\theta }|X)=-\sum_{j}\sum_{i}P(x_{i},y_{j})\log T(\theta_{j}|x_{i})=\bar{d}.$$
*H*(*Y_θ_*) is the semantic entropy:(12)$$H(Y_{\theta })=-\sum_{i}P(y_{j})\log T(\theta_{j}).$$

Note that *P*(*x*|*y_j_*) on the left side of the log is used for averaging and represents the sample distribution. It can be a relative frequency and may not be smooth or continuous. *P*(*x*|*θ_j_*) and *P*(*x*|*y_j_*) may differ, indicating that obtaining information needs factual testing. It is easy to see that the maximum semantic MI criterion is equivalent to the maximum likelihood criterion and similar to the Regularized Least Squares (RLS) criterion. Semantic entropy is the regularization term. Fuzzy entropy is a more general average distortion than the average square error.

Semantic entropy has a clear coding meaning. Assume that the sets *θ*_1_, *θ*_2_, … are crispy sets; the distortion function is(13)$$d(y_{j}|x_{i})=\left\{\begin{array}{c} \infty ,\;x_{i}\notin \theta_{j},\\ 0,\;x_{i}\in \theta_{j}. \end{array}\right.$$
If we regard *P*(*Y*) as the source and *P*(*X*) as the destination, then the parameter solution of the information rate–distortion function is [38](14)$$\begin{array}{l} R(D)=sD(s)-\sum_{j}P(y_{j})\log\lambda_{j},\;\\ \lambda_{j}=\sum_{i}P(x_{i})\exp[-d(x_{i},y_{j})]=\sum_{i}P(x_{i})T(\theta_{j}|x_{i})=T(\theta_{j}). \end{array}$$
It can be seen that the minimum Shannon MI is equal to the semantic entropy, that is, *R*(*D* = 0) = *H*(*Y_θ_*).

The following formula indicates the relationship between the Shannon MI and the semantic MI and the encoding significance of the semantic MI:(15)$$\begin{array}{l} I(X;Y)=\sum_{j}\sum_{i}P(x_{i},y_{j})\log\frac{P(x_{i}|y_{j})}{P(x_{i})}=\\ \sum_{j}\sum_{i}P(x_{i},y_{j})\log\frac{P(x_{i}|\theta_{j})}{P(x_{i})}+\sum_{j}\sum_{i}P(x_{i},y_{j})\log\frac{P(x_{i}|y_{j})}{P(x_{i}|\theta_{j})}\\ =I(X;Y_{\theta })+\sum_{j}P(y_{j})KL(P(x|y_{j})||P(x|\theta_{j})), \end{array}$$
where *KL*(…) is the Kullbak–Leibler (KL) divergence [53] with a likelihood function, which Akaike [54] first used to prove that the minimum KL divergence criterion is equivalent to the maximum likelihood criterion. The last term in the above formula is always greater than 0, reflecting the average code length of the residual coding. Therefore, the semantic MI is less than or equal to the Shannon MI; it indicates the lower limit of the average code length saved due to semantic prediction.

From the above formula, the semantic MI reaches its maximum value when the semantic channel matches the Shannon channel. According to Equation (15), by letting *P*(*x*|*θ_j_*)= *P*(*x*|*y_j_*), we can obtain the optimized truth function from the sample distribution:(16)$$T*(\theta_{j}|x)=\frac{P(x|y_{j})}{P(x)}/\underset{x}{\max}\left(\frac{P(x|y)}{P(x)}\right)=\frac{P(y_{j}|x)}{\underset{x}{\max}(P(y_{j}|x))}.$$
When *Y* = *y_j_*, the semantic MI becomes semantic KL information or semantic side information:(17)$$I(X;\theta_{j})=\sum_{i}P(x_{i}|y_{j})\log\frac{P(x_{i}|\theta_{j})}{P(x_{i})}=\sum_{i}P(x_{i}|y_{j})\log\frac{T(\theta_{j}|x_{i})}{T(\theta_{j})}.$$
The KL divergence cannot usually be interpreted as information because the smaller it is, the better. But the *I*(*X*; *θ_j_*) above can be regarded as information because the larger it is, the better.

Solving *T**(*θ_j_*|*x*) with Equation (16) requires that the sample distributions, *P*(*x*) and *P*(*x*|*y_j_*), are continuous and smooth. Otherwise, by using Equation (17), we can obtain(18)$$T*(\theta_{j}|x)=\underset{\theta_{j}}{\arg\max}\sum_{i}P(x_{i}|y_{j})\log\frac{T(\theta_{j}|x_{i})}{T(\theta_{j})}.$$
The above method for solving *T**(*θ_j_*|*x*) is called Logical Bayes’ Inference (LBI) [7] and can be called the random point falling shadow method. This method inherits Wang’s idea of the random set falling shadow [55,56].

Suppose the truth function in (10) becomes a similarity function. In that case, the semantic MI becomes the estimating MI [32], which has been used by deep learning researchers for Mutual Information Neural Estimation (MINE) [23] and Information Noise Contrast Estimation (InfoNCE) [24].

In the semantic KL information formula, when *X* = *x_i_*, *I*(*X*; *θ_j_*) becomes the semantic information between a single instance *x_i_* and a single label *y_j_*,(19)$$I(x_{i};\theta_{j})=\log\frac{T(\theta_{j}|x_{i})}{T(\theta_{j})}=\log\frac{P(x_{i}|\theta_{j})}{P(x_{i})}.$$

*I*(*x_i_*; *θ_j_*) is called the G measure. This measure reflects Popper’s idea about factual testing. Figure 5 illustrates the above formula. It shows that the smaller the logical probability, the greater the absolute value of the information; the greater the deviation, the less the information; wrong hypotheses convey negative information.

Bringing Equation (2) into (19), we have*I*(*x_i_*; *θ_j_*) = log[1/*T*(*θ_j_*)] − *d*(*y_j_*|*x*), (20)
which means that *I*(*x_i_*; *θ_j_*) equals Carnap and Bar-Hillel’s semantic information minus distortion. 

### 2.5. From the Information Rate–Distortion Function to the Information Rate–Fidelity Function

Shannon defines that given a source, *P*(*x*), a distortion function, *d*(*x*, *y*), and the upper limit, *D*, of the average distortion, $\bar{d}$, we change the channel, *P*(*y*|*x*), to find the minimum MI, *R*(*D*). *R*(*D*) is the information rate–distortion function, which can guide us in using Shannon information economically.

Now, we use *d*(*y_j_*|*x*) = log[1/*T*(*θ_j_*|*x*)] as the asymmetrical distortion function, meaning the distortion when *y_j_* is used as the label of *x*. We replace *d*(*y_j_*|*x_i_*) with *I*(*x_i_*; *θ_j_*), replace $\bar{d}$ with *I*(*X*; *Y_θ_*), and replace *D* with the lower limit, *G*, of the semantic MI to find the minimum Shannon MI, *R*(*G*). *R*(*G*) is the information rate–fidelity function. Because *G* reflects the average codeword length saved due to semantic prediction, using *G* as the constraint is more consistent in shortening the codeword length, and *G*/*R* can represent the information efficiency.

The author uses the word “fidelity” because Shannon originally proposed the information rate–fidelity criterion [1] and later used minimum distortion to express maximum fidelity [38]. The author has previously called *R*(*G*) “the information rate of keeping precision” [6] or “information rate–verisimilitude” [34].

The *R*(*G*) function is defined as(21)$$R(G)=\underset{P(Y|X):I(X;\theta )\geq G}{\min}I(X;Y).$$
We use the Lagrange multiplier method to find the minimum MI and the optimized channel *P*(*y*|*x*). The Lagrangian function is(22)$$L(P(y|x),P(y))=I(X;Y)-sI(X;Y_{\theta })-\mu_{j}\sum_{j}P(y_{j}|x_{i})-\alpha \sum_{j}P(y_{j}).$$

Using *P*(*y*|*x*) as a variation, we let $\partial L/\partial P(y_{j}|x_{i})=0$. Then, we obtain(23)$$P(y_{j}|x_{i})=P(y_{j})m_{ij}^{s}/\lambda_{i},\;\lambda_{i}=\sum_{j}P(y_{j})m_{ij}^{s},i=1,2,\ldots ;j=1,2,\ldots$$
where *m_ij_* = *P*(*x_i_*|*θ_j_*)/*P*(*x_i_*) = *T*(*θ_j_*|*x_i_*)/*T*(*θ_j_*). Using *P*(*y*) as a variation, we let $\partial L/\partial P(y_{j})=0$. Then, we obtain(24)$$P^{+1}(y_{j})=\sum_{i}P(x_{i})P(y_{j}|x_{i}),$$
where *P*^+1^(*y_j_*) means the next *P*(*y_j_*). Because *P*(*y*|*x*) and *P*(*y*) are interdependent, we can first assume a *P*(*y*) and then repeat the above two formulas to obtain the convergent *P*(*y*) and *P*(*y*|*x*) (see [40] (P. 326)). We call this method the Minimum Information Difference (MID) iteration.

The parameter solution of the *R*(*G*) function (as illustrated in Figure 6) is(25)$$\begin{array}{l} G(s)=\sum_{i}\sum_{j}P(x_{i})P(y_{j}|x_{i})I_{ij}=\sum_{i}\sum_{j}I_{ij}P(x_{i})P(y_{j}){m_{ij}}^{s}/Z_{i},\\ R(s)=sG(s)-\sum_{i}P(x_{i})\log Z_{i},{\;Z}_{i}=\sum_{k}P(y_{k}){m_{ij}}^{s}. \end{array}$$

Any *R*(*G*) function is bowl-shaped (possibly not symmetrical [6]), with the second derivative being greater than zero. The *s* = d*R*/d*G* is positive on the right. When *s* = 1, *G* equals *R*, meaning the semantic channel matches the Shannon channel. *G*/*R* represents the information efficiency; its maximum is one. *G* has a maximum value of *G*^+^ and a minimum value of *G*^−^ for a given *R*. *G*^−^ means how small the semantic information the receiver receives can be when the sender intentionally lies.

It is worth noting that, given a semantic channel *T*(*y*|*x*), matching the Shannon channel with the semantic channel, i.e., letting *P*(*y_j_*|*x*) ∝ *T*(*y_j_*|*x*) or *P*(*x*|*y_j_*) = *P*(*x*|*θ_j_*), does not maximize the semantic MI, but minimizes the information difference between *R* and *G* or maximizes the information efficiency, *G*/*R*. Then, we can increase *G* and *R* simultaneously by increasing *s*. When s→∞ in Equation (23), *P*(*y_j_*|*x*) (*j* = 1, 2, …, *n*) only takes the value zero or one, becoming a classification function.

We can also replace the average distortion with fuzzy entropy, *H*(*Y_θ_*|*X*) (using semantic distortion constraints), to obtain the information rate-truth function, *R*(*Θ*) [35]. In situations where information rather than truth is more important, *R*(*G*) is more appropriate than *R*(*D*) and *R*(*Θ*). The *P*(*y*) and *P*(*y*|*x*) obtained for *R*(*Θ*) are different from those obtained for *R*(*G*) because the optimization criteria are different. Under the minimum semantic distortion criterion, *P*(*y*|*x*) becomes(26)$$P(y_{j}|x_{i})=P(y_{j}){\left[T(\theta_{xi}|y_{j})\right]}^{s}/\sum_{j}P(y_{j}){\left[T(\theta_{xi}|y_{j})\right]}^{s},i=1,2,\ldots ;j=1,2,\ldots$$
where *T*(*θ_xi_*|*y*) is a constraint function, so the distortion function *d*(*x_i_*|*y*) = −log *T*(*θ_xi_*|*y*). *R*(*Θ*) becomes *R*(*D*). If *T*(*θ_j_*) is small, the *P*(*y_j_*) required for *R*(*G*) will be larger than the *P*(*y_j_*) required for *R*(*D*) or *R*(*Θ*).

### 2.6. Semantic Channel Capacity

Shannon calls the maximum MI obtained by changing the source, *P*(*x*), for the given Shannon channel, *P*(*y*|*x*), the channel capacity. Because the semantic channel is also inseparable from the Shannon channel, we must provide both the semantic and the Shannon channels to calculate the semantic MI. Therefore, after the semantic channel is given, there are two cases: (1) the Shannon channel is fixed; (2) we must first optimize the Shannon channel using a specific criterion.

When the Shannon channel is fixed, the semantic MI is less than the Shannon MI, so the semantic channel capacity is less than or equal to the Shannon channel capacity. The difference between the two is shown in Equation (15).

If the Shannon channel is variable, we can use the MID iteration to find the Shannon channel for *R* = *G* after each change of the source, *P*(*x*), and then use *s*→∞ to find the Shannon channel, *P*(*y*|*x*), that makes *R* and *G* reach their maxima simultaneously. At this time, *P*(*y*|*x*) ∈ {0, 1} becomes the classification function. Then, we calculate semantic MI. For a different *P*(*x*), the maximum semantic MI is the semantic channel capacity. That is(27)$$C_{T(y|x)}=\underset{P(x);P(y|x)\;\mathrm{for}\;R(G,s\to \infty )}{\arg\max}I(X;Y_{\theta })=\underset{P(x)}{\arg\max}\;G_{\max},$$
where *G*_max_ is *G*^+^ when *s*→∞ (see Figure 6). Hereafter, the semantic channel capacity only means *C_T_*_(*y*|*x*)_ in the above formula.

Practically, to find *C_T_*_(*Y*|*X*)_, we can look for *x*^(1)^, *x*^(2)^, …, *x*^(*j*)^ ∈ ***U***, which are instances under the highest points of *T*(*y*_1_|*x*), *T*(*y*_2_|*x*), …, *T*(*y_n_*|*x*), respectively. Let *P*(*x*^(*j*)^) = 1/*n*, *j* = 1, 2, …, *n*, and the probability of any other *x* equals zero. Then we can choose the Shannon channel: *P*(*y_j_*|*x*^(*j*)^) = 1, *j* = 1, 2, …, *n*. At this time, *I*(*X*; *Y*)) = *H*(*Y*) = log*n*, which is the upper limit of *C_T_*_(*Y*|*X*)_. If there is *x_i_* among the *n x*^(*j*)^*s*, which makes more than one truth function true, then either *T*(*y_j_*) *> P*(*y_j_*) or the fuzzy entropy is not zero. *C_T_*_(*Y*|*X*)_ will be slightly less than log*n* in this case.

According to the above analysis, the encoding method to increase the capacity of the semantic channel is

Try to choose *x* that only makes one label’s truth value close to one (to avoid ambiguity and reduce the logical probability of *y*);Encoding should make *P*(*y_j_*|*x_j_*) = 1 as much as possible (to ensure that *Y* is used correctly).Choose *P*(*x*) so that each *Y*’s probability and logical probability are as equal as possible (close to 1/*n*, thereby maximizing the semantic entropy).

## 3. Electronic Semantic Communication Optimization

### 3.1. The Electronic Semantic Communication Model

The previous discussion of semantic communication did not consider conveying semantic information by electronic communication. Assuming that the time and space distance between the sender and the receiver is very far, we must transmit information through cables or disks. At this time, we need to add an electronic channel to the previous communication model, as shown in Figure 7:

There are two optimization tasks:Task 1: optimizing *P*(*y*|*x*).Task 2: optimizing *P*($\hat{y}$|*y*).

With the optimized *P*(*y*|*x*) and *P*($\hat{y}$|*y*), we can obtain *P*($\hat{x}$|$\hat{y}$) and restore $\hat{x}$ according to *P*($\hat{x}$|$\hat{y}$). The $\hat{x}$ will be quite different from *x*, but the basic feature information will be retained. The restoration quality depends on the feature extraction method.

Task 1 is a machine learning task. Assuming we have selected feature *y*, we can use the previous method to optimize the encoding with semantic information constraints. For example, the overall task is to transmit fruit image information. In Task 1, *x* is a high-dimensional feature vector of the fruit, which contains color and shape information; *y* is the low-dimensional feature vector of a fruit, which contains the fruit variety information. The steps are

(1)Extract the feature vector, *y*, from *x* using machine learning methods (this is a task beyond G theory);(2)Obtain the sample distribution *P*(*y_j_*|*x*) (*j* = 1, 2, …) from the sample;(3)Obtain *T*(*θ_j_*|*x*) ∝ *P*(*y_j_*|*x*) using LBI and further obtain *I*(*x*; *y_j_*) = log[*T*(*θ_j_*|*x*)/*T*(*θ_j_*)] and *G*;(4)Use the method of solving the *R*(*G*) function to obtain the *R* (to ensure that *G* or *G*/*R* is large enough) and the *P*(*y*|*x*) that reflect the encoding rule.

We also need to optimize step (1) according to the classification error between $\hat{x}$ and *x*).

Task 2 is to optimize the electronic channel. Electronic semantic communication is still electronic communication, in essence. The difference is that we need to use semantic information loss instead of distortion as the optimization criterion.

### 3.2. Optimization of Electronic Semantic Communication with Semantic Information Loss as Distortion

Consider electronic semantic communication. If *ŷ_j_* ≠ *y_j_*, there is semantic information loss. Farsad et al. call it a semantic error and propose the corresponding formula [57,58]. Papineni et al. also proposed a similar formula for translation [59]. Gündüz et al. [17] used a method to reduce the loss of semantic information by preserving data features. For more discussion, see [60]. G theory provides the following method for comparison.

According to G theory, the semantic information loss caused by using *ŷ_j_* instead of *y_j_* is(28)$$L_{X}(y_{j}||{\hat{y}}_{j})=I(X;Y_{\theta })-I(X;{\hat{Y}}_{\theta })=\sum_{i}P(x_{i}|y_{j})\log\frac{P(x_{i}|\theta_{j})}{P(x_{i}|{\hat{\theta }}_{j})}.$$
*L_X_*(*y_j_*‖*ŷ_j_*) is a generalized KL divergence because there are three functions. It represents the average codeword length of residual coding.

Since the loss is generally asymmetric, there may be *L_X_*(*y_j_*‖*ŷ_j_*) ≠ *L_X_*(*ŷ_j_*‖*y_j_*). For example, when “motor vehicle” and “car” are substituted for each other, the information loss is asymmetric. The reason is that there is a logical implication relationship between the two. Using “motor vehicle” to replace “car”, although it reduces information, it is not wrong; while using “car” to replace “motor vehicle” may be wrong, because the actual may be a truck or a motorcycle. When an error occurs, the semantic information loss is enormous. An advantage of using the truth function to generate the distortion function is that it can reflect concepts’ implications or similarity relationships.

Assuming *y_j_* is a correctly used label, it comes from sample learning, so *P*(*x*|*θ_j_*) = *P*(*x*|*y_j_*), and *L_X_*(*y_j_*‖*ŷ_j_*) = *KL*(*P*(*x*|*θ_j_*)‖*P*(*x*|$\hat{\theta }$*_j_*)). The average semantic information loss is(29)$$L_{X}(Y||\hat{Y})=\sum_{j}\sum_{k}P(y_{j})P({\hat{y}}_{k}|y_{j})KL(P(x|\theta_{j})||P(x|{\hat{\theta }}_{j})).$$

Consider using *P*(*Y*) as the source and *P*(*Ŷ*) as the destination to encode *y*. Let *d*(*ŷ_k_*|*y_j_*) = *L_X_*(*y_j_*‖*ŷ_k_*); we can obtain the information rate–distortion function *R*(*D*) for replacing *Y* with *Ŷ*. We can code *Y* for data compression according to the parameter solution of the *R*(*D*) function.

In the electronic communication part (from *Y* to *Ŷ*), other problems can be resolved by classical electronic communication methods, except for using semantic information loss as distortion.

If finding *I*(*x*; $\hat{\theta }$*_j_*) is not too difficult, we can also use *I*(*x*; $\hat{\theta }$*_j_*) as a fidelity function. Minimizing *I*(*X*; *ŷ*) for a given *G* = *I*(*X*; *ŷ_θ_*), we can obtain the *R*(*G*) function between *X* and *Ŷ* and compress the data accordingly.

### 3.3. Experimental Results: Compress Image Data According to Visual Discrimination

The simplest visual discrimination is the discrimination of human eyes to different colors or gray levels. The next is the spatial discrimination of points. If the movement of a point on the screen is not perceived, the fuzzy movement range can represent spatial discrimination, which can be represented by a truth function (such as the Gaussian function). What is more complicated is to distinguish whether two figures are the same person. Advanced image compression methods, such as the Autoencoder, need to extract image features and use features to represent images. The following methods need to be combined with the feature extraction methods in deep learning to obtain better applications.

Next, we use the simplest gray-level discrimination as an example to illustrate digital image compression.

(1)Measuring Color Information

A color can be represented by a vector (*B*, *G*, *R*). For convenience, we assume that the color is one-dimensional (or we only consider the gray level), expressed in *x*, and the color sense, *y*, is the estimation of *x*, similar to the GPS indicator. The universes of *x* and *y* are the same, and *y_j_* = “*x* is about *x_j_*”. If the color space is uniform, the distortion function can be defined by distance, that is, *d*(*y_j_*|*x*) = exp[−(*x* − *x_j_*)^2^/(2*σ*^2^)]. Then there is the average information of color perception, *I*(*X*; *Y_θ_*) = *H*(*Y_θ_*) − $\bar{d}$.

Given the source *P*(*x*) and the discrimination function *T*(*y*|*x*), we can solve *P*(*y*|*x*) and *P*(*y*) using the Semantic Variational Basyes (SVB) method [61]. The Shannon channel is matched with the semantic channel to maximize the information efficiency.

(2)Gray Level Compression

We used an example to demonstrate color data compression. It was assumed that the original gray level was 256 (8-bit pixels) and needed to be compressed into 8 (3-bit pixels). We defined eight constraint functions, as shown in Figure 8a.

Considering that human visual discrimination varies with the gray level (the higher the gray level, the lower the discrimination), we used the eight truth functions shown in Figure 8a, representing eight fuzzy ranges. Appendix C in Reference [37] shows how these curves are generated. The task was to use the maximum information efficiency (MIE) criterion to find the Shannon channel *P*(*y*|*x*) that made *R* close to *G* (*s* = 1).

The convergent *P*(*y*|*x*) is shown in Figure 8b. Figure 8c shows that the Shannon MI and the semantic MI gradually approach in the iteration process. Comparing Figure 8a,b, we find it easy to control *P*(*y*|*x*) by *T*(*y*|*x*). However, defining the distortion function without using the truth function is difficult. Predicting the convergent *P*(*y*|*x*) by *d*(*y*|*x*) is also difficult.

If we use *s* to strengthen the constraint, we obtain the parametric solution of the *R*(*G*) function. As *s*→∞, *P*(*y_j_*|*x*) (*j* = 1, 2, …) display as rectangles and become classification functions.

(3)The Influence of Visual Discrimination and the Quantization Level on the *R*(*G*) Function

The author performed some experiments to examine the influence of the discrimination function and the quantization level *b* = 2*^n^* (*n* is the number of quantization bits) on the *R*(*G*) function. Figure 9 shows that when the quantization level was enough, the *R* and *G* variation range increased with the discrimination increasing (i.e., with *σ* decreasing). The result explains how the discrimination determines the semantic channel capacity.

Figure 9a indicates that higher discrimination can convey more semantic information for a given quantization level (*b* = 63). Figure 9b shows that for a given discrimination (*σ* = 1/64), less *b* will waste the semantic channel capacity. For more discussion on visual information, see Section 6 in [6].

## 4. Goal-Oriented Information, Information Value, Physical Entropy, and Free Energy

### 4.1. Three Kinds of Information Related to Value

We call the increment of utility the value. Information involves utility and value in three aspects:

**(1) Information about utility.** For example, the information about university admissions or the bumper harvest of grain is about utility. The measurement of this information is the same as in the previous examples. Before providing information, we have the prior probability distribution, *P*(*x*), of grain production. The information is provided by ranges, such as “about 2000 kg per acre”, which can be expressed by a truth function. The previous semantic information formula is also applicable.

**(2) Goal-oriented information** [61]. This is also purposeful information or constraint control feedback information.

For example, a passenger and a driver watch a GPS map in a taxi. Assume that the probability distribution of the taxi position without looking at the positioning map (or without some control) is *P*(*x*), and the destination is a fuzzy range, which is represented by a truth function. The actual position is the probability distribution, *P*(*x*|*a_j_*) (*a_j_* is an action). The positioning map provides information. For the passenger, this is purposeful information (about how the control result comforts the purpose); for the driver, this is the control feedback information. We call both types goal-oriented information. This information involves constraint control and reinforcement learning. Section 3.2 discusses the measurement and optimization of this information.

**(3) Information that brings value.** For example, Tom made money by buying stocks based on John’s prediction of stock prices. The information provided by John brings Tom increased utility, so John’s information is valuable to Tom.

The value of information is relative. For example, weather forecast information is different for workers and farmers, and prediction information about stock markets is worth zero to people who do not buy stocks. Information value is often difficult to judge. For example, value losses due to missed reporting and false reporting are difficult regarding medical cancer tests. In most cases, the missed reporting of low-probability events often causes more loss than false reporting, such as medical tests and earthquake forecasts. In these cases, the semantic information criterion can be used to reduce the missed reporting of low-probability events.

For investment portfolios, the quantitative analysis of the information value is possible. Section 4.3 focuses on the information values of portfolios.

### 4.2. Goal-Oriented Information

#### 4.2.1. Similarities and Differences Between Goal-Oriented Information and Prediction Information

Previously, we used the G measure to measure prediction information, requiring the prediction, *P*(*x*|*θ_j_*), to conform to the fact, *P*(*x*|*y_j_*). Goal-oriented information is the opposite, requiring the fact to conform to the purpose.

An imperative sentence can be regarded as a control instruction. We need to know whether the control result conforms to the control purpose. The more consistent the result is, the more information there is.

A truth function or a membership function can represent a control target. For example, there are the following targets:“Workers’ wages should preferably exceed 5000 dollars”;“The age of death of the population had better exceed 80 years old”;“The cruising distances of electric vehicles should preferably exceed 500 km”;“The error of train arrival time had better be less than one minute”.

The semantic KL information formula can measure purposeful information:(30)$$I(X;a_{j}/\theta_{j})=\sum_{i}P(x_{i}|a_{j})\log\frac{T(\theta_{j}|x_{i})}{T(\theta_{j})}.$$

In the formula, *θ_j_* is a fuzzy set, indicating that the control target is a fuzzy range. Here, *y_j_* becomes *a_j_*, indicating the action corresponding to the *j*-th control task, *y_j_*. If the control result is a specific *x_i_*, the above formula becomes the semantic information *I*(*x_i_*; *a_j_*/*θ_j_*).

If there are several control targets, *y*_1_, *y*_2_, …, we can use the semantic MI formula to express the purposeful information:(31)$$I(X;A/\theta )=\sum_{j}P(a_{j})\sum_{i}P(x_{i}|a_{j})\log\frac{T(\theta_{j}|x_{i})}{T(\theta_{j})},$$
where *A* is a random variable taking a value *a* or *a_j_*. Using SVB, the control ratio, *P*(*a*), can be optimized to minimize the control complexity (i.e., Shannon MI) for given fuzzy range constraints.

#### 4.2.2. The Optimization of Goal-Oriented Information

Goal-oriented information can be regarded as the cumulative reward in constraint control. However, the goal here is a fuzzy range, which is expressed by a plan, command, or imperative sentence. The optimization task is similar to the active inference task using the MFE principle [22].

The semantic information formulas of imperative and descriptive (or predictive) sentences are the same, but the optimization methods differ (see Figure 10). For descriptive sentences, the fact is unchanged, and we hope that the predicted range conforms to the fact, that is, we fix *P*(*y_j_*|*x*) so that *T*(*θ_j_*|*x*) ∝ *P*(*y_j_*|*x*), or we fix *P*(*x*|*y_j_*) so that *P*(*x*|*θ_j_*) = *P*(*x*|*y_j_*). For imperative sentences, we hope that the fact conforms to the purpose, that is, we fix *T*(*θ_j_*|*x*) or *P*(*x*|*θ_j_*) and minimize the information difference or maximize the information efficiency *G*/*R* by changing *P*(*y_j_*_|_*x*) or *P*(*x*|*y_j_*), or we balance between the purposiveness and the efficiency.

For multi-target tasks, the objective function to be minimized is*f* = *I*(*X*; *A*) − *sI*(*X*; *A*/*θ*).(32)

When the actual distribution, *P*(*x*|*a_j_*), is close to the constrained distribution, *P*(*x*|*θ_j_*), the information efficiency (not information) reaches its maximum value of one. To further increase the two types of information, we can use the MID iteration formula to obtain(33)$$P(a_{j}|x)=P(a_{j})m_{ij}^{s}/\lambda_{i},$$(34)$$P(x_{i}|a_{j})=P(a_{j}|x_{i})P(x_{i})/P(a_{j})=P(x_{i})m_{ij}^{s}/\sum_{k}P(x_{k})m_{kj}^{s}.$$

Compared with VB [20,27,28,29] for active inference, the above method is simpler and can change the constraint strength by *s*.

Because the optimized *P*(*x*|*a_j_*) is a function of *θ_j_* and *s*, we write *P**(*x*|*a_j_*) = *P*(*x*|*θ_j_*, *s*). It is worth noting that many distributions, *P*(*x*|*a_j_*), satisfy the constraint and maximize *I*(*X*; *a_j_*/*θ_j_*), but only *P**(*x*|*a_j_*) minimizes *I*(*X*; *a_j_*).

#### 4.2.3. Experimental Results: Trade-Off Between Maximizing Purposiveness and MIE

Figure 11 shows a two-objective control task, with the objectives represented by the truth functions *T*(*θ*_0_|*x*) and *T*(*θ*_1_|*x*). We can imagine these as two pastures with fuzzy boundaries where we need to herd sheep. Without control, the density distribution of the sheep is *P*(*x*). We need to solve an appropriate distribution, *P*(*a*).

For a different *s*, we set the initial proportions: *P*(*a*_0_) = *P*(*a*_1_) = 0.5. Then, we used Equations (23) and (24) for the MID iteration to obtain the proper *P*(*a_j_*|*x*) (*j* = 0, 1). Then, we obtained *P*(*x*|*a_j_*) = *P*(*x*|*θ_j_*,*s*) by using Equation (34). Finally, we obtained *G*(*s*), *R*(*s*), and *R*(*G*) by using Equation (25).

The dashed line for *R*_1_(*G*) in Figure 12 indicates that if we replace *P*(*x*|*a_j_*) = *P*(*x*|*θ_j_*, *s*) with a normal distribution, *P*(*x*|*β_j_*, *s*), *G*, and *G*/*R*_1_ do not obviously become worse.

### 4.3. Investment Portfolios and Information Values

#### 4.3.1. Capital Growth Entropy

Markowitz’s portfolio theory [62] uses a linear combination of expected income and standard deviation as the optimization criterion. In contrast, the compound interest theory of portfolios uses compound interest, i.e., geometric mean income, as the optimization criterion.

The compound interest theory began from Kelley [63], followed by Latanne and Tuttle [64], Arrow [65], Cover [66], and the author of this paper. The famous American information theory textbook “*Elements of Information Theory*” [67], co-authored by Cover and Thomas, introduced Cover’s research. Arrow, Cover, and the author of this article also discussed information value. The author published a monograph, “*Entropy Theory of Portfolios and Information Value*” [68] in 1997 and obtained many different conclusions.

The following is a brief introduction to the capital growth entropy proposed by the author.

Assuming that the principal is *A*, the profit is *B*, and the sum of principal plus profit is *C*. The investment income is *r* = *B*/*A*, and the rate of return on investment (i.e., output ratio: output/input) is *R* = *C*/*A* = 1 + *r*.

*N* security prices form an *N*-dimensional vector, and the price of the *k*-th security has *n_k_* possible prices, *k* = 1, 2, …, *N*. There are *W* = *n*_1_ × *n*_2_ × … × *n_N_* possible price vectors. The *i*-th price vector is *x_i_* = (*x_i_*_1_, *x_i_*_2_, … *x_iN_*), *i* = 1, 2, …, *W*; the current price vector is *x*_0_ = (*x*_01_, *x*_02_, …, *x*_0*N*_). Assuming that one year later, the price vector *x_i_* occurs, then the rate of return of the *k*-th security is *R_ik_* = *x_ik_*/*x*_0*k*_, and the total rate of return is(35)$$R_{i}=\sum_{k=0}^{N}q_{k}R_{ik}$$
where *q_k_* is the investment proportion in the *k*-th security, *q*_0_ is the proportion of cash (or risk-free assets) held by the investor. There is *R*_0*k*_ = *R*_0_ = (1 + *r*_0_), where *r*_0_ is the risk-free interest rate.

Suppose we conduct *m* investment experiments to obtain the price vectors, and the number of times *x_i_* or *r_i_* occurs is *m_i_*. The average multiple of the capital growth after each investment period or the geometric mean output ratio is(36)$$R_{g}=\prod_{i=1}^{W}{R_{i}}^{m_{i}/m}.$$

When *m*→∞, we have *m_i_*/*m* = *P*(*x_i_*) and the capital growth entropy(37)$$H_{g}=\log R_{g}=\sum_{i=1}^{W}P(x_{i})\log R_{i}=\sum_{i=1}^{W}P(x_{i})\log\sum_{k=0}^{N}q_{k}R_{ik}.$$

If the log is base 2, *H_g_* represents the doubling rate.

If the investment turns into betting on horse racing, where only one horse (the *k*-th horse) wins each time, the winner’s return rate is *R_k_*, and the others lose their wagers. Then, the above formula becomes(38)$$H_{g}=\log R_{g}=\sum_{k}P(x_{k})\log[q_{0}+q_{k}R_{k}-(1-q_{0}-q_{k})],$$
where *q*_0_ is the proportion of funds not betted, and 1 − *q*_0_ − *q*_k_ is the proportion of funds paid.

#### 4.3.2. The Generalization of Kelley’s Formula

Kelley, a colleague of Shannon, found that the method used by Shannon’s information theory can be used to optimize betting, so he proposed the Kelley formula [63].

Assume that in a gambling game, if you lose, you will lose *r*_1_ = 1 times; if you win, you will earn *r*_2_ > 0 times. If the probability of winning is *P*, then the optimal ratio is*q** = *P* − (1 − *P*)/*r*_2_. (39)

Using capital growth entropy can lead to more general conclusions. Let *r*_1_ < 0. The capital growth entropy is(40)$$q*=\underset{q}{\arg\max}\;H_{g}=P_{1}\log(1-qr_{1})+P_{2}\log(1+qr_{2}).$$

Letting d*H_g_*/d*q* = 0, we derive*q** = *E*/(*r*_1_*r*_2_), (41)
where *E* is the expected income. For example, for a coin toss bet, if one wins, he earns twice as much; if he loses, he loses one times; the probabilities of winning and losing are equal. Therefore, *E* is 0.5, and the optimal investment ratio is *q** = 0.5/(1 × 2) = 0.25.

We assume *r*_0_ = 0 above. If we consider the opportunity cost or the risk-free income, then *r*_0_ > 0. At this time, the optimal ratio is(42)$$\begin{array}{ll} q* & =\underset{q}{\arg\max}\;H_{g}\\ & =\underset{q}{\arg\max}\{P_{1}\log[r_{0}(1-q)+q-qr_{1}]+P_{2}\log[r_{0}(1-q)+q+qr_{2}]\}. \end{array}$$

Letting d*H_g_*/d*q* = 0, we can obtain(43)$$q*=\frac{P_{2}d_{2}-P_{1}d_{1}}{d_{1}d_{2}}R_{0},$$
where *R*_0_ = 1 + *r*_0_, *d*_1_ = *r*_1_ + *r*_0_, and *d*_2_ = *r*_2_ − *r*_0_.

The author’s abovementioned book [68] also discusses optimizing the investment ratio when short selling and leverage are allowed (see Section 3.3 in [68]) and optimizing the investment ratio for multi-coin betting. The book derives the limit theorem of diversified investment: if the number of coins increases infinitely, the geometric mean income equals the arithmetic mean income.

#### 4.3.3. Risk Measures, Investment Channels, and Investment Channel Capacity

Markowitz uses the expected income, *E*, and the standard deviation, *σ*, to represent the income and the risk of a portfolio. Similarly, we use *R_g_* and *R_r_* to represent the return and risk of a portfolio. *R_r_* is defined in the following formula:(44)$$R_{r}^{2}=R_{a}^{2}-R_{g}^{2},$$
where *R_a_* = 1 + *E*. Assuming that the geometric mean return of any portfolio is equivalent to the geometric mean return of a coin toss bet with an equal probability of gain or loss, then*H_g_* = log*R_g_* = 0.5log(*R_a_* − *R_r_*) + 0.5log(*R_a_* + *R_r_*). (45)

Let sinα = *R_r_*/*R_a_* ∈ [0, 1], which represents the bankruptcy risk better. When sinα is close to one, the investment may go bankrupt (see Figure 13).

We call the pair (***P***, ***R***) the investment channel, where ***P*** = (*P*_1_, *P*_2_, … *P_M_*) is the future price vector, ***R*** = (*R_ik_*) is the return matrix, and the set of all possible investment ratio vectors is ***q**_C_*. Then, the capacity of the investment channel (abbreviated as investment capacity) is defined as(46)$$H_{C}*=\underset{q\in q_{C}}{\max}\;H(P,R,q)=H(P,R,q*),$$
where ***q**** = ***q****(***R***, ***P***) is the optimal investment ratio.

For example, for a typical coin toss bet (with equal probabilities of winning and losing, and *r*_0_ = 0), *q** = *E*/(*r*_1_*r*_2_), the investment capacity is(47)$$H_{C}*=\frac{1}{2}\log\frac{1}{1-E^{2}/{R_{r}}^{2}}.$$

Since 1/(1 − *x*) = 1 + *x* + *x*^2^ + … ≈ 1 + *x*, when *E*/*R_r_* << 1, there is an approximate formula:(48)$$H_{C}*\approx \frac{1}{2}\log(1+\frac{E^{2}}{{R_{r}}^{2}}).$$

In comparison with the Gaussian channel capacity formula for communication,(49)$$C=\frac{1}{2}\log(1+\frac{P}{N}),$$
we can see that the investment capacity formula is very similar to the Gaussian channel capacity formula. This similarity means that investment needs to reduce risk, just as communication needs to reduce noise.

#### 4.3.4. The Information Value Formula Based on Capital Growth Entropy

Weaver, who co-authored the book “*A Mathematical Theory of Communication*” [2] with Shannon, proposed three communication levels related to Shannon’s information, semantic information, and information value.

According to the common usage of “information value”, the information value mentioned in the academic community does not refer to the value of information on markets but to the utility or utility increment generated by information. We define the information value as the increment of capital growth entropy.

Assume that the prior probability distribution of different returns is *P*(*x*), and the return matrix is (*R_ik_*), then the expected capital growth entropy is *H_g_*(*X*). The optimal investment ratio vector ***q**** is ***q**** = ***q****(*P*(*x*), (*R_ik_*)). When the probability distribution of the predicted return becomes *P*(*x*|*θ_j_*), the capital growth entropy becomes(50)$$H_{g}*(X|\theta_{j})=\sum_{i}P(x_{i}|\theta_{j})\log R_{i}(q*),R_{i}(q*)=\sum_{k}q_{k}*R_{ik}.$$

The optimal investment ratio becomes ***q***** = ***q*****(*P*(*x*|*θ_j_*), (*R_ik_*)). We define the increment of the capital growth entropy due to the semantic KL information *I*(*X*; *θ_j_*) as the information value(51)$$V(X;\theta_{j})=\sum_{i}P(x_{i}|y_{j})\log\frac{R_{i}(q**)}{R_{i}(q*)}.$$

*V*(*X*; *θ_j_*) and *I*(*X*; *θ_j_*) have similar structures. For the above formula, when *x_i_* is determined to occur, the information value of *y_j_* becomes(52)$$v_{ij}=v(x_{i};\theta_{j})=\log\frac{R_{i}(q**)}{R_{i}(q*)}.$$

The information value also needs to be verified by facts; wrong predictions may bring a negative information value. In contrast, Cover and Thomas [67] do not distinguish *P*(*x*|*θ_j_*) and *P*(*x*|*y_j_*).

#### 4.3.5. Comparison with Arrow’s Information Value Formula

The utility function defined by Arrow [65] is(53)$$U=\sum_{i}P_{i}U(q_{i}R_{i})=\sum_{i}P_{i}\log(q_{i}R_{i})=\sum_{i}P_{i}\log q_{i}+\sum_{i}P_{i}\log R_{i},$$
where *U*(*q_i_R_i_*) is the utility obtained by the investor when the *i*-th return occurs.

Under the restriction of ∑*_i_ q_i_* = 1, *q_i_* = *P_i_* (*i* = 1, 2, …) maximizes *U* so that(54)$$U*=\sum P_{i}\log P_{i}+\sum_{i}P_{i}\log R_{i}.$$

After receiving the information, the investor knows which income will occur and thus invests all his funds in it. Hence, there is(55)$$U**=\sum_{i}P_{i}\log R_{i}.$$

The information value is defined as the difference in utility between investment with and without information and is equal to Shannon entropy, that is(56)$$V=U**-U*=-\sum_{i}P_{i}\log P_{i}=H(X).$$

The optimal investment ratio obtained from the above formula is inconsistent with the Kelley formula and common sense. For example, according to the Kelley formula, the optimal ratio is 25% for the coin toss bet above. According to common sense, if we know the result of the coin toss, the return will depend on the odds.

According to Arrow’s theory, profit and loss have nothing to do with odds. How does one bet? Should one bet by using *q_i_* = *P_i_* or by putting all in the profit? Arrow seems to confuse the *k*-th security with the *i*-th return. He uses *U*(*q_i_R_i_*) = log(*q_i_R_i_*), while the author uses(57)$$U(q_{k}R_{k})=\log[q_{0}+q_{k}R_{k}-(1-q_{0}-q_{k})].$$

Arrow does not consider the non-bet proportion, *q*_0_, nor the paid proportion, 1 − *q*_0_ − *q_k_*. Calculating the utility in this way is puzzling.

Cover and Thomas inherited Arrow’s method and concluded that when there is information, the optimal investment doubling rate increment equals the Shannon MI [67] (see Section 6.2). Their conclusion has the same problem.

### 4.4. Information, Entropy, and Free Energy in Thermodynamic Systems

To clarify the relationship between information and free energy in physics, we discuss information, entropy, and free energy in thermodynamic systems.

Jaynes [69] used Stirling’s formula, ln*N*! = *N*ln*N* − *N* (when *N*→∞), to prove there is a simple connection between Boltzmann’s microscopic sate number *W* (of *N* molecules) and Shannon entropy:(58)$$S=k\ln W=k\ln\frac{N!}{\prod_{i}N_{i}!}=-kN\sum_{i}P(x_{i}|T)\ln P(x_{i}|T)=kNH(X|T),$$
where *S* is entropy, k is the Boltzmann constant, *N* is the number of molecules, *x_i_* is the *i*-th microscopic state of one molecule, and *T* is the absolute temperature, which equals a molecule’s average translational kinetic energy. *P*(*x_i_*|*T*) represents the probability density of molecules in state *x_i_* at temperature *T*. The Boltzmann distribution for a given energy constraint is(59)$$P(x_{i}|T)=\exp(-\frac{e_{i}}{kT})/Z^{\prime },Z^{\prime }=\sum_{i}\exp(-\frac{e_{i}}{kT}),$$
where *Z*′ is the partition function.

Considering the information between temperature and molecular energy, we use Maxwell–Boltzmann statistics. In this case, we replace *x_i_* with energy *e_i_* and let *G_i_* denote the number of microscopic states of one molecule with energy *e_i_* and let *G* denote the number of all states of one molecule. Then *P*(*e_i_*) = *G_i_*/*G* is the prior probability of *e_i_*. So, Equation (58) becomes(60)$$\begin{array}{ll} S & =k\ln\frac{N!}{\prod_{i}N_{i}!/{G_{i}}^{N_{i}}}=-kN\sum_{i}P(e_{i}|T)\ln\frac{P(e_{i}|T)}{G_{i}}\\ & =-kN\sum_{i}P(e_{i}|T)\ln\frac{P(e_{i}|T)}{P(e_{i})}+kN\ln G\\ & =kN[\ln G-KL(P(e|T)||P(e)]. \end{array}$$
Under the energy constraint, when the system reaches equilibrium, Equation (59) becomes(61)$$P(e_{i}|T)=P(e_{i})\exp(-\frac{e_{i}}{kT})/Z,Z=\sum_{i}P(e_{i})\exp(-\frac{e_{i}}{kT}),$$

Now, we can interpret exp[−*e_i_*/(*kT*)] as the truth function *T*(*θ_j_*|*x*), *Z* as the logical probability *T*(*θ_j_*), and Equation (61) as the semantic Bayes’ formula.

Consider a local non-equilibrium system. Different regions *y_j_* (*j* = 1, 2, …) of the system have different temperatures, *T_j_* (*j* = 1, 2, …). Hence, we have *P*(*y_j_*) = *P*(*T_j_*) = *N_j_*/*N* and *P*(*x*|*y_j_*) = *P*(*x*|*T_j_*). From Equation (60), we obtain (62)$$\begin{array}{cc} I(E;Y) & =\sum_{j}P(y_{j})KL(P(e|y_{j})||P(e)=\sum_{j}P(y_{j})[\ln G_{m}-S_{j}/(kN_{j})]\\ & =\ln G_{m}-S(kN), \end{array}$$
where *E* is a random variable taking a value *e*, and ln*G_m_* is the prior entropy *H*(*X*) of *X*. Since *e* is certain for a given *x*, there are *H*(*E*|*X*) = 0 and *H*(*X*, *E*) = *H*(*X*) = ln*G_m_*. From Equations (60) and (62), we derive(63)$$I(E;Y)=\ln G_{m}-S/(kN)=H(X)-H(X|Y)=I(X;Y).$$

This formula indicates that the information about *E* provided by *Y* or *T* is equal to the information about *X*, and *S*/(*kN*) equals *H*(*X*|*Y*). The above formula shows that the entropy increase law in physics can be equivalently expressed as the MI decrease law.

According to Equations (61)–(63), when the local equilibrium is reached, there is(64)$$\begin{array}{l} \;I(X;Y)=I(E;Y)=\sum_{j}\sum_{i}P(e_{i},y_{j})\ln\frac{\exp[-e_{i}/(kT_{j})]}{Z_{j}}\\ =-\sum_{j}P(y_{j})\log Z_{j}-E(e/T)/(kN)=H(Y_{\theta })-H(Y_{\theta }|X)=I(X;Y_{\theta }), \end{array}$$
where E(*e*/*T*) is the average of *e*/*T*, which is similar to relative square error. It can be seen that in local equilibrium systems, minimum Shannon MI can be expressed by the semantic MI formula. Since *H*(*X*|*Y*) becomes *H*(*X*|*Y_θ_*), there is(65)$$\;S=kNH(X|Y_{\theta })=kNF,$$
which means that VFE, *F*, is proportional to thermodynamic entropy.

Helmholtz’s free energy formula is*F** = *U* − *TS*,(66)
where *F** is free energy, and *U* is the system’s internal energy. When the internal energy remains unchanged, the increase in free energy is(67)$$\;\Delta F*=-\Delta (TS)=TS-\sum_{j}T_{j}S_{j}=kNTH(X)-kN\sum_{j}T_{j}H(X|Y).$$

Comparing the above equation with Equation (63), we can find that the Shannon MI is analogous to the increase in free energy; the semantic MI is analogous to the increase in free energy in a local equilibrium system, which is smaller than the Shannon MI, just as work is smaller than costed free energy. We can also regard k*NT* and k*NT_j_* as the unit information values [5], so the increase in free energy is equivalent to the increase in the information value.

## 5. G Theory’s Applications in Machine Learning

### 5.1. Basic Methods of Machine Learning: Learning Functions and Optimization Criteria

The most basic machine learning method has two steps:First, we use samples or sample distributions to train the learning functions with a specific criterion, such as the maximum likelihood or RLS criterion;Then, we make probability predictions or classifications utilizing the learning function with the minimum distortion, minimum loss, or maximum likelihood criterion.

We use the maximum likelihood or RLS criterion when training learning functions. We may use different criteria when using learning functions for classifications. We generally use maximum likelihood and RLS criteria for prediction tasks where information is important. To judge whether a person is guilty or not, where correctness is essential, we may use the minimum distortion (or loss) criterion. The maximum semantic information criterion is equivalent to the maximum likelihood criterion and is a Regularized Least Distortion (RLD) criterion, including the partition function’s logarithm. Compared with the minimum distortion criterion, the maximum semantic information criterion can reduce the under-reporting of small probability events.

We generally do not use *P*(*x*|*y_j_*) to train *P*(*x*|*θ_j_*) because if *P*(*x*) changes, the originally trained *P*(*x*|*θ_j_*) will become invalid. A parameterized transition probability function, *P*(*θ_j_*|*x*), as a learning function, is still valid even if *P*(*x*) is changed. However, using *P*(*θ_j_*|*x*) as a learning function also has essential defects. When class number *n* > 2, it is challenging to construct *P*(*θ_j_*|*x*) (*j* = 1, 2, …) because of the normalization restriction, that is, ∑*_j_ P*(*θ_j_*|*x*) = 1 (for each *x*). As we will see below, there is no restriction when using truth or membership functions as learning functions.

The following sections introduce G theory’s applications to machine learning. If we use existing methods, every task is not easy. Either the solution is complicated, or the convergence proof is difficult.

### 5.2. For Multi-Label Learning and Classification

Consider multi-label learning, a supervised learning task. From the sample {(*xk*, *yk*), *k* = 1, 2, …, *N*}, we can obtain the sample distribution, *P*(*x*, *y*). Then, we use Formula (16) or (18) for the optimized truth functions.

Assume that a truth function is a Gaussian function, there should be(68)$$T(\theta_{j}|x)\propto \frac{P(x|y_{j})}{P(x)}\propto P(y_{j}|x).$$

So, we can use the expectation and standard deviation of *P*(*x*|*y_j_*)/*P*(*x*) or *P*(*y_j_*|*x*) as the expectation and the standard deviation of *T*(*θ_j_*|*x*). If the truth function is like a dam cross-section (see Figure 14), we can obtain it through some transformation.

If we only know *P*(*y_j_*|*x*) but not *P*(*x*), we can assume that *P*(*x*) is equally probable, that is, *P*(*x*) = 1/|***U***|, and then optimize the membership function using the following formula:(69)$$I(X;\theta_{j})=\sum_{i}P(x_{i}|y_{j})\log\frac{T(\theta_{j}|x_{i})}{T(\theta_{j})}=\sum_{i}\frac{P(y_{j}|x_{i})}{\sum_{k}P(y_{j}|x_{k})}\log\frac{T(\theta_{j}|x_{i})}{\sum_{k}T(\theta_{j}|x_{k})}+\log|U|.$$

For multi-label classification, we can use the classifier(70)$$y_{j}*=\underset{y_{j}}{\arg\max}\;I(x;\theta_{j})=\underset{y_{j}}{\arg\max}\;\log\frac{T(\theta_{j}|x)}{T(\theta_{j})}.$$

If the distortion criterion is used, we can use −log *T*(*θ_j_*|*x*) as the distortion function or replace *I*(*X*; *θ_j_*) with *T*(*θ_j_*|*x*).

The popular Binary Relevance [70] for multi-label classification converts an *n*-label learning task into an *n*-pair label learning task. In comparison, the above method is much simpler.

### 5.3. Maximum MI Classification of Unseen Instances

This classification belongs to semi-supervised learning. We take the medical test and the signal detection as examples (see Figure 15).

The following algorithm is not limited to binary classifications. Let *C_j_* be a subset of *C* and *y_j_* = *f*(*z*|*z* ∈ *C_j_*); hence, *S* = {*C*_1_, *C*_2_, …} is a partition of *C*. Our task is to find the optimal *S*, which is(71)$$S*=\underset{S}{\arg\max}\;I(X;Y_{\theta }|S)=\underset{S}{\arg\max}\sum_{j}\sum_{i}P(C_{j})P(x_{i}|C_{j})\log\frac{T(\theta_{j}|x_{i})}{T(\theta_{j})}.$$

First, we initiate a partition. Then, we do the following iterations.

**Matching I:** Let the semantic channel match the Shannon channel and set the reward function. First, for given *S*, we obtain the Shannon channel:(72)$$P(y_{j}|x)=\sum_{z_{k}\in C_{j}}P(z_{k}|x),\;j=1,2,\ldots ,n.$$

Then we obtain the semantic channel, *T*(*y*|*x*), from the Shannon channel and *T*(*θ_j_*) (or *m_θ_*(*x*, *y*) = *m*(*x*, *y*)). Then we have *I*(*x_i_*; *θ_j_*). For a given *z*, we have conditional information as the reward function:(73)$$I(X;\theta_{j}|z)=\sum_{i}P(x_{i}|z)I(x_{i};\theta_{j}),\;j=0,1,\ldots ,n,$$

**Matching II:** Let the Shannon channel match the semantic channel by the classifier(74)$$y_{j}*=f(z)=\underset{y_{j}}{\arg\max}\;I(X;\theta_{j}|z),\;j=0,1,\ldots ,n.$$

Repeat **Matching I** and **Matching II** until *S* does not change. Then, the convergent *S* is the *S** we seek. The author explained the convergence with the *R*(*G*) function (see Section 3.3 in [7]).

Figure 6 shows an example. The detailed data can be found in Section 4.2 of [7]. The two lines in Figure 16a represent the initial partition. Figure 16d shows that the convergence is very fast.

However, this method is unsuitable for the maximum MI classification in a high-dimensional space. We need to combine neural network methods to explore more effective approaches.

### 5.4. The Explanation and Improvement of the EM Algorithm for Mixture Models

The EM algorithm [71,72] is usually used for mixure models (clustering), an unsupervised learning method.

We know that *P*(*x*) = ∑*_j_ P*(*y_j_*)*P*(*x*|*y_j_*). Given a sample distribution, *P*(*x*), we use *P_θ_*(*x*) = ∑*_j_ P*(*y_j_*)*P*(*x*|*θ_j_*) to approximate *P*(*x*) so that the relative entropy or KL divergence, *KL*(*P*‖*P_θ_*), is close to zero. *P*(*y*) is the probability distribution of the latent variable to be sought.

The EM algorithm first presets *P*(*x*|*θ_j_*) and *P*(*y_j_*), *j* = 1, 2, …, *n*. The E-step obtains(75)$$P(y_{j}|x)=P(y_{j})P(x|\theta_{j})/P_{\theta }(x),\;P_{\theta }(x)=\sum_{k}P(y_{k})P(x|\theta_{k}).$$

Then, in the M-step, the log-likelihood of the complete data (usually represented by *Q*) is maximized. The M-step can be divided into two steps: M1-step for(76)$$P^{+1}(y_{j})=\sum_{i}P(x_{i})P(y_{j}|x_{i})$$
and M2-step for(77)$$P(x|{\theta_{j}}^{+1})=P(x)P(y_{j}|x)/P^{+1}(y_{j})=P(x)\frac{P(x|\theta_{j})}{P_{\theta }(x)}\frac{P(y_{j})}{P^{+1}(y_{j})},$$
which optimizes the likelihood function. For Gaussian mixture models, we can use the expectation and standard deviation of *P*(*x*)*P*(*y_j_*|*x*)/*P*^+1^(*y_j_*) as the expectation and standard deviation of *P*(*x*|*θ_j_*^+1^).

From the perspective of G theory, the M2-step is to make the semantic channel match the Shannon channel or minimize the VFE, *H*(*X*|*Y_θ_*), and the E-step and M1-step are to match the Shannon channel with the semantic channel. Repeating the above three steps can make the mixture model converge. The converged *P*(*y*) is the required probability distribution of the latent variable. According to the derivation process of the *R*(*G*) function, the E-step and M1-step minimize the information difference, *R*–*G*; the M-step maximizes the semantic MI. Therefore, the optimization criterion used by the EM algorithm is the MIE criterion.

However, there are two problems with the above method to find the latent variable: (1) *P*(*y*) may converge slowly; (2) if the likelihood functions are also fixed, how do we solve *P*(*y*)?

Based on the *R*(*G*) function analysis, the author improved the EM algorithm to the EnM algorithm [7]. The EnM algorithm includes the E-step for *P*(*y*|*x*), the n-step for *P*(*y*), and the M-step for *P*(*x*|*θ_j_*) (*j* = 1, 2, …). The n-step repeats the E-step and the M1-step in the EM algorithm *n* times so that *P*^+1^(*y*) ≈ *P*(*y*). The EnM algorithm also uses the MIE criterion. The n-step can speed up the solution of *P*(*y*). The M2-step only optimizes the likelihood functions. Because *P*(*y_j_*)/*P*^+1^(*y_j_*) is approximately equal to one, we can use the following formula to optimize the following model parameters:(78)$$P(x|{\theta_{j}}^{+1})=P(x)P(x|\theta_{j})/P_{\theta }(x).$$
Without the n-step, there will be *P*(*y_j_*) ≠ *P*^+1^(*y_j_*), and ∑ *_i_ P*(*x_i_*)*P*(*x*|*θ_j_*)/*P_θ_*(*x_i_*) ≠ 1. When solving mixure models, we can choose a smaller *n*, such as *n* = 3. When solving *P*(*y*) specifically, we can select a larger *n* until *P*(*y*) converges. When *n* = 1, the EnM algorithm becomes the EM algorithm.

The following mathematical formula proves that the EnM algorithm converges. After the M-step, the Shannon MI becomes(79)$$R=\sum_{i}\sum_{j}P(x_{i})\frac{P(x_{i}|\theta_{j})}{P_{\theta }(x_{i})}P(y_{j})\log\frac{P(y_{j}|x_{i})}{P^{+1}(y_{j})},$$

We define(80)$$R^{\prime \prime }=\sum_{i}\sum_{j}P(x_{i})\frac{P(x_{i}|\theta_{j})}{P_{\theta }(x_{i})}P(y_{j})\log\frac{P(x_{i}|\theta_{j})}{P_{\theta }(x_{i})}.$$
Then, we can deduce that after the E-step, there is(81)$$KL(P||P_{\theta })=R^{\prime \prime }-G=R-G+KL({P_{Y}}^{+1}\left|\right|P_{Y}),$$
where *KL*(*P*‖*P_θ_*) is the relative entropy or KL divergence between *P*(*x*) and *P_θ_*(*x*); the right KL divergence is(82)$$KL({P_{Y}}^{+1}\left|\right|P_{Y})=\sum_{j}P^{+1}(y_{j})\log[P^{+1}(y_{j})/P(y_{j})].$$
It is close to zero after the n-step.

Equation (81) can be used to prove that the EnM algorithm converges. Because the M-step maximizes *G*, and the E-step and the n-step minimize *R*–*G* and *KL*(*P_Y_*^+1^‖*P_Y_*), *H*(*P*‖*P_θ_*) can be close to zero. We can also use the above method to prove that the EM algorithm converges.

In most cases, the EnM algorithm performs better than the EM algorithm, especially when *P*(*y*) is hard to converge.

Some researchers believe that EM makes the mixture model converge because the complete data log-likelihood *Q* = −*H*(*X*, *Y_θ_*) continues to increase [72], or the negative free energy *F*′ = *H*(*Y*) + *Q* continues to increase [21]. However, we can easily find counterexamples where *R*–*G* continues to decrease, but *Q* and *F*′ do not necessarily continue to increase.

The author used the example used by Neal and Hinton [21] (see Figure 17), but the mixture proportion in the true model was changed from 0.7:0.3 to 0.3:0.7.

This experiment shows that the decrease in *R*–*G*, not the increase in *Q* or *F*′, is the reason for the convergence of the mixture model.

The free energy of the true mixture model (with true parameters) is the Shannon conditional entropy *H*(*X*|*Y*). If the standard deviation of the true mixture components is large, *H*(*X*|*Y*) is also large. If the initial standard deviation is small, *F* is small initially. After the mixture model converges, *F* must approach *H*(*X*|*Y*). Therefore, *F* increases (i.e., *F*′ decreases) during the convergence process. For example, a true mixture model with two components with two standard deviations of 15. If two initial standard deviations are 5, *F* must continue increasing during the iteration. Many experiments have shown that this is indeed the case.

Equation (81) can also explain pre-training in deep learning, where we need to maximize the model’s predictive ability and minimize the information difference, *R*–*G* (or compress data).

### 5.5. Semantic Variational Bayes: A Simple Method for Solving Hidden Variables

Given *P*(*x*) and constraints *P*(*x*|*θ_j_*), *j* = 1, 2, …, we need to solve *P*(*y*) that produces *P*(*x*) = ∑*_j_ P*(*y_j_*)*P*(*x*|*θ_j_*). *P*(*y*) is the probability distribution of the latent variable *y*, sometimes called the latent variable. The popular method is the Variational Bayes method (VB for short) [27,28,29]. This method originated from the article by Hinton and Camp [20]. It was further discussed and applied in the articles by Neal and Hinton [21], Beal [28], and Koller [29] (ch. 11). Gottwald and Braun’s article, “Two Free Energy and the Bayesian Revolution” [73], discusses the relationship between the MFE principle and Maximum Entropy (ME) principle in detail.

VB uses *P*(*y*) (usually written as *g*(*y*)) as a variation to minimize the following function:(83)$$F=\sum_{y}g(y)\log\frac{g(y)}{P(x,y|\theta )}=-\sum_{y}g(y)\log P(x|y,\theta )+KL(g(y)||P(y))$$
Using the semantic information method, we express *F* as(84)$$F=\sum_{i}P(x_{i})\sum_{i}P(y_{j}|x_{i})\log\frac{P^{+1}(y_{j})}{P(x_{i},y_{j}|\theta )}=H(X|Y_{\theta })+KL({P_{Y}}^{+1}||P_{Y}).$$

Since *F* is equal to the semantic posterior entropy *H*(*X*|*Y_θ_*) of *X* after the M1-step of the EM algorithm, we can treat *F* as *H*(*X*|*Y_θ_*). Since *I*(*X*; *Y_θ_*)) = *H*(*X*) − *H*(*X*|*Y_θ_*) = *F* − *H*(*X*|*Y*), the smaller the *F*, the larger the semantic MI.

It is easy to prove that when the semantic channel matches the Shannon channel, that is, *T*(*θ_j_*|*x*) ∝ *P*(*y_j_*|*x*) or *P*(*x*|*θ_j_*) = *P*(*x*|*y_j_*) (*j* = 1, 2, …), *F* is minimized, and the semantic MI is maximized. Minimizing *F* can optimize the prediction model, *P*(*x*|*θ_j_*) (*j* = 1, 2, …), but it cannot optimize *P*(*y*). For optimizing *P*(*y*), the mean-field approximation [27,28] is used; that is, *P*(*y*|*x*) instead of *P*(*y*) is used as the variation. Only one *P*(*y_j_*|*x*) is optimized at a time, and the other *P*(*y_k_*|*x*) (*k* ≠ *j*) remains unchanged. Minimizing *F* in this way is actually maximizing the log-likelihood of *x* or minimizing *KL*(*P*‖*P_θ_*). In this way, optimizing *P*(*y*|*x*) also indirectly optimizes *P*(*y*).

Unfortunately, when optimizing *P*(*y*) and *P*(*y*|*x*), *F* may not continue to decrease (see Figure 17). So, VB is suitable as a tool but imperfect as a theory.

Fortunately, it is easier to solve *P*(*y*|*x*) and *P*(*y*) using the MID iteration in solving *R*(*D*) and *R*(*G*) functions. The MID iteration plus LBI for optimizing the prediction model is SVB [61]. It uses the MIE criterion.

When the constraint changes from likelihood functions to truth functions or similarity functions, *P*(*y_j_*|*x_i_*) in the MID iteration formula is changed from(85)$$P(y|x_{i})=P(y){\left[\frac{P(x_{i}|\theta_{j})}{P_{\theta }(x_{i})}\right]}^{s}/\sum_{k}P(y_{k}){\left[\frac{P(x_{i}|\theta_{j})}{P_{\theta }(x_{i})}\right]}^{s}$$
to(86)$$P(y|x_{i})=P(y){\left[\frac{T(\theta_{j}|x_{i})}{T(\theta_{j})}\right]}^{s}/\sum_{k}P(y_{k}){\left[\frac{T(\theta_{k}|x_{i})}{T(\theta_{k})}\right]}^{s}.$$

From *P*(*x*) and the new *P*(*y*|*x*), we can obtain the new *P*(*y*). Repeating the formulas for *P*(*y*|*x*) and *P*(*y*) will lead to the convergence of *P*(*y*). Using *s* allows us to tighten the constraints for increasing *R* and *G*. Choosing proper *s* enables us to balance between maximizing semantic information and maximizing information efficiency.

The main tasks of SVB and VB are the same: using variational methods to solve latent variables according to observed data and constraints. The differences are

**Criteria**: In the definition of VB, it adopts the MFE criterion, whereas, for solving *P*(*y*), it uses *P*(*y*|*x*) as the variation and hence actually uses the maximum likelihood criterion that makes the mixture model converge. In contrast, SVB uses the MID criterion.**Variational method**: VB only uses *P*(*y*) or *P*(*y*|*x*) as a variation, while SVB alternatively uses *P*(*y*|*x*) and *P*(*y*) as variations.**Computational complexity**: VB uses logarithmic and exponential functions to solve *P*(*y*|*x*) [27]; the calculation of *P*(*y*|*x*) in SVB is relatively simple (for the same task, i.e., when *s* = 1).**Constraints**: VB only uses likelihood functions as constraint functions. In contrast, SVB allows using various learning functions (including likelihood, truth, membership, similarity, and distortion functions) as constraints. In addition, SVB can use the parameter *s* to enhance constraints.

Because SVB is more compatible with the maximum likelihood criterion and the ME principle, it should be more suitable for many applications in machine learning. However, because it does not consider the probability of parameters, it may not be as applicable as VB on some occasions.

### 5.6. Bayesian Confirmation and Causal Confirmation

Logical empiricism was opposed by Popper’s falsificationism [44,45], so it turned to confirmation (i.e., Bayesian confirmation) instead of induction or positivism [74,75]. Bayesian confirmation was previously a field of concern for researchers in the philosophy of science [76,77], and now many researchers in natural sciences have also begun to study it [78,79]. The reason is that uncertain reasoning requires major premises, which need to be confirmed.

The main reasons why researchers have different views on Bayesian confirmation are

There are no suitable mathematical tools; for example, statistical and logical probabilities are not well distinguished.Many people do not distinguish between the confirmation of the relationship (i.e., →) in the major premise *y*→*x* and the confirmation of the consequent (i.e., *x* occurs);No confirmation measure can reasonably clarify the raven paradox [74].

To clarify the raven paradox, the author wrote the article “Channels’ confirmation and predictions’ confirmation: from medical tests to the Raven paradox” [35].

In the author’s opinion, the task of Bayesian confirmation is to evaluate the support of the sample distribution for the major premise. For example, for the medical test (see Figure 15), a major premise is “If a person tests positive (*y*_1_), then he is infected (*x*_1_)”, abbreviated as *y*_1_→*x*_1_. For a channel’s confirmation, a truth (or membership) function can be regarded as a combination of a clear truth function, *T*(*y*_1_|*x*) ∈ {0,1}, and a tautology’s truth function (always one):*T*(*θ*_1_|*x*) = *b*_1_*T*(*y*_1_|*x*) + *b*_1_′.(87)

A tautology’s proportion, *b*_1_′, is the degree of disbelief. The credibility is *b*_1_, and its relationship with *b*_1_′ is *b*_1_′ = 1 − |*b*_1_|. See Figure 18.

We change *b*_1_ to maximize the semantic KL information, *I*(*X*; *θ*_1_); the optimized *b*_1_, denoted as *b*_1_*, is the confirmation degree:(88)$$b_{1}*=b*(y_{1}\to x_{1})=\frac{P(y_{1}|x_{1})-P(y_{1}|x_{0})}{\max(P(y_{1}|x_{1}),P(y_{1}|x_{0}))}=\frac{R^{+}-1}{\max(R^{+},1)},$$
where *R*^+^ =*P*(*y*_1_|*x_i_*)/*P*(*y*_1_|*x*_0_) is the positive likelihood ratio, indicating the reliability of the tested positive. This conclusion is compatible with medical test theory.

Considering the prediction confirmation degree, we assume that *P*(*x*|*θ*_1_) is a combination of the 0–1 part and the probability-equal part. The ratio of the 0–1 part is the prediction credibility, and the optimized credibility is the prediction confirmation degree:(89)$$c_{1}*=c*(y_{1}\to x_{1})=\frac{P(x_{1}|y_{1})-P(x_{0}|y_{1})}{\max(P(x_{1}|y_{1}),P(x_{0},y_{1}))}=\frac{a-c}{\max(a,c)},$$
where *a* is the number of positive examples, and *c* is the number of negative examples.

Both confirmation degrees can be used for probability predictions, i.e., calculating *P*(*x*|*θ*_1_).

Hemple proposed the confirmation paradox, namely the raven paradox [74]. According to the equivalence condition in classical logic, “If *x* is a raven, then *x* is black” (Rule 1) is equivalent to “If *x* is not black, then *x* is not a raven” (Rule 2). According to this, white chalk supports Rule 2; therefore, it also supports Rule 1. However, according to common sense, a black crow supports Rule 1, and a non-black raven opposes Rule 1; something that is not a raven, such as a black cat or a white chalk, is irrelevant to Rule 1. Therefore, there is a paradox between the equivalence condition and common sense. Using the confirmation measure *c*_1_*, we can ensure that common sense is correct and the equivalence condition for fuzzy major premises is wrong, thus eliminating the raven paradox. However, other confirmation measures cannot eliminate the raven paradox [35].

Causal probability is used in causal inference theory [80]:(90)$$P_{d}=\max[0,\frac{P(y_{1}|x_{1})-P(y_{1}|x_{0})}{P(y_{1}|x_{1})}]=\max(0,\frac{R^{+}-1}{R^{+}}).$$

It indicates the necessity of the cause, *x*_1_, replacing *x*_0_ to lead to the result, *y*_1_, where *P*(*y*_1_|*x*) = *P*(*y*_1_|do(*x*)) is the posterior probability of *y*_1_ caused by intervention *x*. The author uses the semantic information method to obtain the channel causal confirmation degree [36]:(91)$$Cc(x_{1}/x_{0}=>y_{1})=b_{1}*=\frac{P(y_{1}|x_{1})-P(y_{1}|x_{0})}{\max(P(y_{1}|x_{1}),P(y_{1}|x_{0}))}=\frac{R^{+}-1}{\max(R^{+},1)}.$$

It is compatible with the above causal probability but can express negative causal relationships, such as the necessity of vaccines inhibiting infection because it can be negative.

### 5.7. Emerging and Potential Applications

(1)About self-supervised learning.

Applications of estimating MI have emerged in the field of self-supervised learning. The estimating MI is a special case of semantic MI. Both MINE, proposed by Belghazi et al. [23], and InfoNCE, proposed by Oord et al. [24], use the estimating MI. MINE and InfoNCE are essentially the same as the semantic information methods. Their common features are

The truth function, *T*(*θ_j_*|*x*), or similarity function, *S*(*x*, *y_j_*), proportional to *P*(*y_j_*|*x*) is used as the learning function. Its maximum value is generally one, and its average is the partition function, *Z_j_*.The estimating information or semantic information between *x* and *y_j_* is log[*T*(*θ_j_*|*x*)/*Z_j_*] or log[*S*(*x*, *y_j_*)/*Z_j_*].The statistical probability distribution, *P*(*x*, *y*), is still used when calculating the average information.

However, many researchers are still unclear about the relationship between the estimating MI and the Shannon MI. G theory’s *R*(*G*) function can help readers understand this relationship.

(2)About reinforcement learning.

The goal-oriented information introduced in Section 4.2 can be used as a reward for reinforcement learning. Assuming that the probability distribution of *x* in state *s_k_* is *P*(*x*|*a_k_*_−1_), which becomes *P*(*x*|*a_k_*) in state *s_k_*_+1_, the reward of *a_k_* is(92)$$r_{k}=I(X;a_{k}/\theta_{j})-I(X;a_{k-1}/\theta_{j})=\sum_{i}\left[P(x_{i}|a_{k})-P(x_{i}|a_{k-1})\right]\log\frac{T(\theta_{j}|x_{i})}{T(\theta_{j})}.$$

Reinforcement learning is to find the optimal action sequence *a*_1_, *a*_2_, …, so that the sum of rewards *r*_1_ + *r*_2_ + … is maximized. Like constraint control, reinforcement learning also needs the trade-off between the maximum purposefulness and the minimum control cost. The *R*(*G*) function should be helpful.

(3)About the truth function and fuzzy logic for neural networks.

When we use the truth, distortion, or similarity function as the weight parameter of the neural network, the neural network contains semantic channels. Then, we can use semantic information methods to optimize the neural network. Using the truth function, *T*(*θ_j_*|*x*), as the weight is better than using the parameterized inverse probability function, *P*(*θ_j_*|*x*), because there is no normalization restriction when using truth functions.

However, unlike the clustering of points on the plane, points become images for image clustering, and the similarity function between images needs different methods. A common method is to regard an image as a vector and use the cosine similarity between vectors. However, the cosine similarity may have negative values, which require activation functions and biases to make necessary conversions. Combining existing neural network methods and channel-matching algorithms needs further exploration.

Fuzzy logic, especially fuzzy logic compatible with Boolean algebra [52], seems to be useful in neural networks; for example, the activation function, *Relu*(*a*–*b*) = max(0, *a*–*b*), which is commonly used in neural networks, is the logical difference operation, *f*(*a*$\bar{b}$) = max(0, *a*–*b*), used in the author’s color vision mechanism model. Truth functions, fuzzy logic, and the semantic information method used in neural networks should make neural networks easier to understand.

(4)Explaining data compression in deep learning.

To explain the success of deep neural networks, such as AutoEncoders [33] and Deep Belief Networks [81], Tishby et al. [31] proposed the information bottleneck explanation, affirming that when optimizing deep neural networks, we maximize the Shannon MI between some layers and minimize the Shannon MI between other layers. However, from the perspective of the *R*(*G*) function, each coding layer of the Autoencoder needs to maximize the semantic MI and minimize the Shannon MI; pre-training is to let the semantic channel match the Shannon channel so that *G* ≈ *R* and *KL*(*P*‖*P_θ_*) ≈ 0 (as if for mixture models to converge). Fine-tuning increases *R* and *G* at the same time by increasing *s* (making the partition boundaries steeper).

Not long ago, researchers at OpenAI [82,83] explained General Artificial Intelligence by lossless (actually, loss-limited) data compression, similar to the explanation of using MIE.

## 6. Discussion and Summary

### 6.1. Core Idea and Key Methods of Generalizing Shannon’s Information Theory

In order to overcome the three shortcomings of Shannon’s information theory (it is not suitable for semantic communication, lacks the information criterion, and cannot bring model parameters or likelihood functions into the MI formula), the core idea for the generalization is to replace distortion constraints with semantic constraints. Semantic constraints include semantic distortion constraints (using log[1/*T*(*θ_j_*|*x*)] as the distortion function), semantic information quantity constraints, and semantic information loss constraints (for electronic semantic communication). In this way, the shortcomings can be overcome, and existing coding methods can be used.

One key method is using the P-T probability framework with the truth function. The truth function can represent semantics, form a semantic channel, and link likelihood and distortion functions so that sample distributions can be used to optimize learning functions (likelihood function, truth function, similarity function, etc.).

The second key method is to define semantic information as the negative regularized distortion. In this way, the semantic MI equals the semantic entropy minus the average distortion, i.e., *I*(*X*; *Y_θ_*) = *H_θ_*(*Y*) − $\bar{d}$. The MI between temperature and molecular energy also has this form in a thermodynamic local equilibrium system. The logarithm of the softmax function, which is widely used in machine learning, also has this form [37]. This regularization’s characteristic is that the term has the form of semantic entropy, which contains the logarithm of the logical probability, *T*(*θ_j_*), or the partition function, *Z_j_.* We call it “partition regularization” to distinguish it from other forms of regularization. Why can this semantic information measure also measure the information in local equilibrium systems? The reason is that the semantic constraint is similar to the energy constraint; both are fuzzy range constraints and can be represented by negative exponential functions.

Because the MFE criterion is equivalent to the Partition-Regularized Least Distortion (PRLD) criterion, the successful applications of VB and the MFE principle also support the PRLD criterion.

Because *T*(*θ_j_*) represents the average of the true value, log[*T*(*θ_j_*|*x*)/*T*(*θ_j_*)] represents the progress from not so true to true. Comparing the core part of the Shannon MI, log[*P*(*x*|*y_j_*)/*P*(*x*)], we can say that Shannon information comes from the improvement of probability. In contrast, semantic information comes from the improvement of truth. Furthermore, because log[*T*(*θ_j_*|*x*)/*T*(*θ_j_*)] = log[*P*(*x*|*θ_j_*)/*P*(*x*)], the PRLD criterion is equivalent to the maximum likelihood criterion. Machine learning researchers always believe that regularization can reduce overfitting. Now, we find that, more importantly, it is compatible with the maximum likelihood criterion.

Because of the above two key methods, the three shortcomings of Shannon information theory no longer exist in G theory. Moreover, the success of VB and the MFE principle prompted Shannon information theory researchers to use the minimum VFE criterion or the maximum semantic information criterion to minimize the residual coding length of predictive coding.

Semantic constraints include semantic information loss constraints for electronic communication (see Section 3.2). Using this constraint, we can use classic coding methods to achieve electronic semantic communication.

### 6.2. Some Views That Are Different from Those G Theory Holds

**View 1**: We can measure the semantic information of language itself.

Some people want to measure the semantic information provided by a sentence (such as Carnap and Bar-Hillel), and others measure the semantic information between two sentences, regardless of the facts. In the author’s opinion, these practices ignore the source of information: the real world. G theory follows Popper’s idea and affirms that information comes from factual testing; if a hypothesis conforms to the facts and has a small prior logical probability, there is more information; if it is wrong, there is less or negative information. Some people may say that the translation between two sentences transmits semantic information. In this regard, we must distinguish between actual translation and the formulation of translation rules. Actual translation does provide semantic information, while translation rules do not provide semantic information, but they determine semantic information loss. Therefore, the actual translation should use the maximum semantic information criterion, while optimizing translation rules should use the minimum semantic information loss criterion. Optimizing electronic semantic communication is similar to optimizing translation rules, and the minimum semantic information loss criterion should be used.

**View 2**: Semantic communication needs to transmit semantics.

Because the transmission of Shannon information requires a physical channel, some people believe that the transmission of semantic information also requires a corresponding physical channel or utilizes the physical channel to transmit semantics. In fact, generally speaking, the semantic channel already exists in the brain or knowledge of the sender and the receiver, and there is no need to consider the physical channel. Only when the knowledge of both parties is inconsistent do we need to consider such a physical channel. The picture–text dictionary is a good physical channel for transmitting semantics. However, the receiver only needs to look at it once, and it will not be needed in the future.

**View 3**: Semantic information also needs to be minimized.

In Shannon’s information theory, the MI, *R*, should be minimized when the distortion limit, *D*, is given to improve communication efficiency. The information rate–distortion function, *R*(*D*), provides the theoretical basis for data compression. Therefore, some people imitated the information rate–distortion function and proposed the semantic information rate–distortion function, *R_s_*(*D*), where *R_s_* is the minimum semantic MI. However, from the perspective of G theory, although we consider semantic communication, the communication cost is still Shannon’s MI, which needs to be minimized. Therefore, G theory replaces the average distortion with the semantic MI to evaluate communication quality and uses the *R*(*G*) function.

**View 4**: Semantic information measures do not require encoding meaning.

Some people believe that the concept of semantic information has nothing to do with encoding and that constructing a semantic information measure does not require considering its encoding meaning. However, the author holds the opposite view. The reasons are (1) semantic information is related to uncertainty, and thus also to encoding; (2) many successful machine learning methods have used cross-entropy to reflect the encoding length and VFE (i.e., posterior cross-entropy *H*(*X*|*Y_θ_*)) to indicate the lower limit of residual encoding length or reconstruction cost.

The G measure is equal to *H*(*X*) − *F*, reflecting the encoding length saved because of semantic predictions. The encoding length is an objective standard, and a semantic information measure without an objective standard makes it hard to avoid subjectivity and arbitrariness.

### 6.3. What Is Information? Is the G Measure Compatible with the Daily Information Concept?

Is the G measure, a technical information measure, compatible with the daily information concept? This cannot be ignored.

What is information? This question has many different answers, as summarized by Mark Burgin [84]. According to Shannon’s definition, information is uncertainty reduced. Shannon information is the uncertainty reduced due to the increase of probability. In contrast, semantic information is the uncertainty reduced due to narrowing concepts’ extensions or improving truth.

From a common-sense perspective, information refers to something previously unknown or uncertain, which encompasses

(1)**Information from natural language:** information provided by answers to interrogative sentences (e.g., sentences with “Who?”, “What?”, “When?”, “Where?”, “Why?”, “How?”, or “Is this?”).(2)**Perceptual or observational information:** information obtained from material properties or observed representations.(3)**Symbolic information:** information conveyed by symbols like road signs, traffic lights, and battery polarity symbols [12].(4)**Quantitative indicators’ information: information provided by** data, such as time, temperature, rainfall, stock market indices, inflation rates, etc.(5)**Associated information:** information derived from event associations, such as the rooster’s crowing signaling dawn and a positive medical test indicating disease.

Items 2, 3, and 4 can also be viewed as answers to the questions in item 1, thus providing information. These forms of information involve concept extensions and truth–falsehood and should be semantic information. The associated information in item 5 can be measured using Shannon’s or semantic information formulas. When probability predictions are inaccurate (i.e., *P*(*x*|*θ_j_*) ≠ *P*(*x*|*y_j_*)), the semantic information formula is more appropriate. Thus, G theory is consistent with the concept of information in everyday life.

In computer science, information is often defined as useful, structured data. What qualifies as “useful”? This utility arises because the data can answer questions or provide associated information. Therefore, the definition of information in data science also ties back to reduced uncertainty and narrowed concept extensions.

### 6.4. Relationships and Differences Between G Theory and Other Semantic Information Theories

#### 6.4.1. Carnap and Bar-Hillel’s Semantic Information Theory

The semantic information measure of Carnap and Bar-Hillel is [3]*Ip* = log(1/*m_p_*], (93)
where *I_p_* is the semantic information provided by the proposition set, *p*, and *m_p_* is the logical probability of *p*. This formula reflects Popper’s idea that smaller logical probabilities convey more information. However, as Popper noted, this idea requires the hypothesis to withstand factual testing. The above formula does not account for such tests, implying that correct and incorrect hypotheses provide the same information.

Additionally, G theory differs in calculating logical probability with statistical probability, unlike Carnap and Bar-Hillel’s approach.

#### 6.4.2. Dretske’s Knowledge and Information Theory

Dretske [9] emphasized the relationship between information and knowledge, viewing information as content tied to facts and knowledge acquisition. Though he did not propose a specific formula, his ideas about information quantification include

The information must correspond to facts and eliminate all other possibilities.The amount of information relates to the extent of the uncertainty eliminated.Information used to gain knowledge must be true and accurate.

G theory is compatible with these principles by providing the G measure to implement Dretske’s idea mathematically.

#### 6.4.3. Florida’s Strong Semantic Information Theory

Florida’s theory [12] emphasizes

The information must contain semantic content and be consistent with reality.False or misleading information cannot qualify as true information.

Floridi elaborated on Dretske’s ideas and introduced a strong semantic information formula. However, this formula is more complex and less effective at reflecting factual testing than the G measure. For instance, Floridi’s approach ensures that tautologies and contradictions yield zero information but fails to penalize false predictions with negative information.

#### 6.4.4. Other Semantic Information Theories

In addition to the semantic information theories mentioned above, other well-known ones include the theory based on fuzzy entropy proposed by Zhong [10] and the theory based on synonymous mapping proposed by Niu and Zhang [16]. Zhong advocated for the combination of information science and artificial intelligence, which greatly influenced China’s semantic information theory research. He employed fuzzy entropy [85] to define the semantic information measure. However, this approach yielded identical maximum values (1 bit) for both true and false sentences [10], which is not what we expect. Other people’s semantic information measures using DeLuca and Termini’s fuzzy entropy also encounter similar problems.

Other authors who discussed semantic information measures and semantic entropy include D’Alfonso [13], Basu et al. [86], and Melamed [87]. These authors improved semantic information measures by improving Carnap and Bar-Hillel’s logical probability. The form of semantic entropy is similar to the form of Shannon entropy. The semantic entropy, *H*(*Y_θ_*), in G theory differs from these semantic entropies. It contains statistical and logical probabilities and reflects the average codeword length of lossless coding (see Section 2.3).

Liu et al. [88] and Guo et al. [89] used the dual-constrained rate distortion function, *R*(*D_s_*, *D_x_*), which is meaningful. In contrast, *R*(*G*) already contains dual constraints because the semantic constraints include distortion and semantic information constraints.

In addition, some fuzzy information theories [90,91] and generalized information theories [92] also involve semantics more or less. However, most of them are far away from Shannon’s information theory.

### 6.5. Relationship Between G Theory and Kolmogorov’s Complexity Theory

Kolmogorov [93] defined the complexity of a set of data as the shortest program length required to restore the data without loss:(94)K(data)=min{|p| |U(p)=data}.
where *U*(*p*) is the output of program *p*. It defines the information provided by knowledge as the complexity reduced due to knowledge [84].

The information measured by Shannon can be understood as the average information provided by *y* about *x*, while Kolmgorov’s information is the information provided by knowledge about the co-occurring data. Shannon’s information theory does not consider the complexity of an individual datum, while Kolmogorov’s theory does not consider statistical averages. It can be said that Kolmogorov defines the amount of information in microdata, while Shannon provides a formula to measure the average information in macrodata. The two theories are complementary.

The semantic information measure, *I*(*X*; *Y_θ_*), is related to Kolmogorov complexity in the following two aspects:Because knowledge includes the extensions of concepts, the logical relationship between concepts, and the correlation between things (including causality), the information Kolmogorov said contains semantic information.Because VFE means the reconstruction cost due to the prediction, and the G measure equals *H*(*X*) minus VFE, the G measure is similar to Kolmogorov’s information measure. Both mean the saved reconstruction cost.

Some people have proposed complexity distortion [94], the shortest coding length with an error limit. It is possible to extend complexity distortion to the shortest coding length with the semantic constraint to obtain a function similar to the *R*(*G*) function. This direction is worth exploring.

### 6.6. Comparing the MIE Principle and the MFE Principle

Friston proposed the MFE principle, which he believed was a universal principle that bio-organisms use to perceive the world and adapt to the environment (including transforming the environment). The core mathematical method he uses is VB. A better-understood principle in G theory is the MIE principle.

The main differences between the two are

G theory regards Shannon’s MI *I*(*X*; *Y*) as free energy’s increment, while Friston’s theory considers the semantic posterior entropy, *H*(*X*|*Y_θ_*), as free energy.The methods for finding the latent variable, *P*(*y*), and the Shannon channel, *P*(*y*|*x*), are different. Friston uses VB, and G theory uses SVB.

When optimizing the prediction model, *P*(*x*|*θ_j_*) (*j* = 1, 2, …), the two are consistent; when optimizing *P*(*y*) and *P*(*y*|*x*), the results of the two are similar, but the methods are different. SVB is simpler. The reason why the results are similar is that VB uses the mean-field approximation when optimizing *P*(*y*|*x*), which is equivalent to using *P*(*y*|*x*) instead of *P*(*y*) as a variation and actually uses the MID criterion. Figure 18 shows that the information difference, *R*–*G*, instead of VFE continuously decreases in a mixture model’s convergence process.

In physics, free energy can be used to do work; the more, the better. Why should it be minimized? In physics, there are two situations in which free energy is reduced. One is passive reduction because of the increase in entropy. The other reason is to save the consumed free energy while doing work due to considering thermal efficiency. Reducing the consumed free energy conforms to Jaynes’ maximum entropy principle. Therefore, from a physics perspective, it is not easy to understand that one would actively minimize free energy.

MIE is like the maximum doing-work efficiency, *W*/*ΔF**, when using free energy to perform work, *W*. The MIE principle is easier to understand.

The author will discuss these two principles further in other articles.

### 6.7. Limitations and Areas That Need Exploration

G theory is still a basic theory. It has limitations in many aspects, which need improvements.

(1)Semantics and the distortion of complex data.

Truth functions can represent the semantic distortion of labels. However, it is difficult to express the semantics, semantic similarity, and semantic distortion of complex data (such as a sentence or an image). Many researchers have made valuable explorations [14,15,17,95,96,97]. The author’s research is insufficient.

The semantic relationship between a word and many other words is also very complex. Innovations like Word2Vec [98,99] in deep learning have successfully modeled these relationships, paving the way for advancements like Transformer [100] and ChatGPT. Future work in G theory should aim to integrate such developments to align with the progress in deep learning.

(2)Feature extraction.

The features of images encapsulate most semantic information. There are many efficient feature extraction methods in deep learning, such as Convolutional Neural Networks [101] and AutoEncoders [33]. These methods are ahead of G theory. Whether G theory can be combined with these methods to obtain better results needs further exploration.

(3)The channel-matching algorithm for neural networks.

Establishing neural networks to enable the mutual alignment of Shannon and semantic channels appears feasible. Current deep learning practices, relying on gradient descent and backpropagation, demand significant computational resources. If the channel-matching algorithm can reduce reliance on these methods, it would save computational power and physical free energy.

(4)Neural networks utilizing fuzzy logic.

Using truth functions or their logarithms as network weights facilitates the application of fuzzy logic. The author previously used a fuzzy logic method [52] compatible with Boolean algebra for setting up a symmetrical color vision mechanism model: the 3–8 decoding model. Combining G theory with fuzzy logic and neural networks holds promise for further exploration.

(5)Optimizing economic indicators and forecasts.

Forecasting in weather, economics, and healthcare domains provides valuable semantic information. Traditional evaluation metrics, such as accuracy or the average error, can be enhanced by converting distortion into fidelity. Using truth functions to represent semantic information offers a novel method for evaluating and optimizing forecasts, which merits further exploration.

### 6.8. Conclusions

The semantic information G theory is a generalized Shannon’s information theory. Its validity has been supported by its broad applications across multiple domains, particularly in solving problems related to daily semantic communication, electronic semantic communications, machine learning, Bayesian confirmation, constraint control, and investment portfolios with information values.

Particularly, G theory allows us to utilize the existing coding methods for semantic communication. The core idea is to replace distortion constraints with semantic constraints, for which the information rate–distortion function was extended to the information rate–fidelity function. The key methods are (1) to use the P-T probability framework with truth functions and (2) to define the semantic information measure as the negative partition-regularized distortion.

However, G theory’s primary limitation lies in the semantic representation of complex data. In this regard, it has lagged behind the advancements in deep learning. Bridging this gap will require learning from and integrating insights from other studies and technologies.

## Figures and Tables

**Figure 1 entropy-27-00461-f001:**
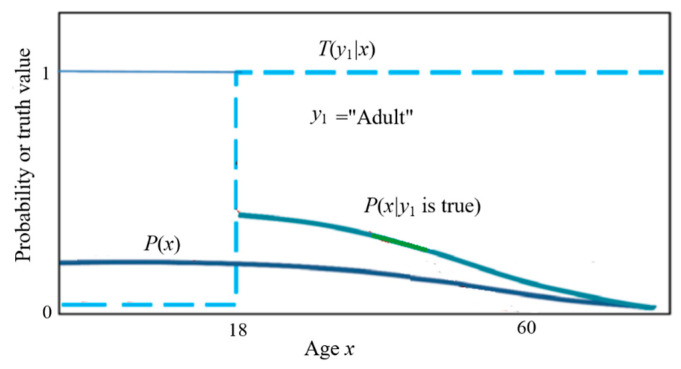
The semantic probability prediction, according to that “*x* is adult” is true.

**Figure 2 entropy-27-00461-f002:**
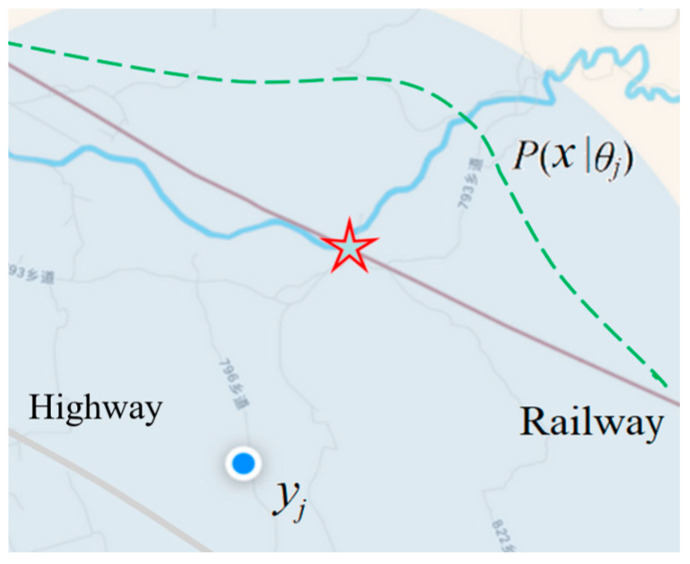
A GPS device’s positioning with a deviation. The round point is the pointed position with a deviation, and the place with the star is the most likely position.

**Figure 3 entropy-27-00461-f003:**
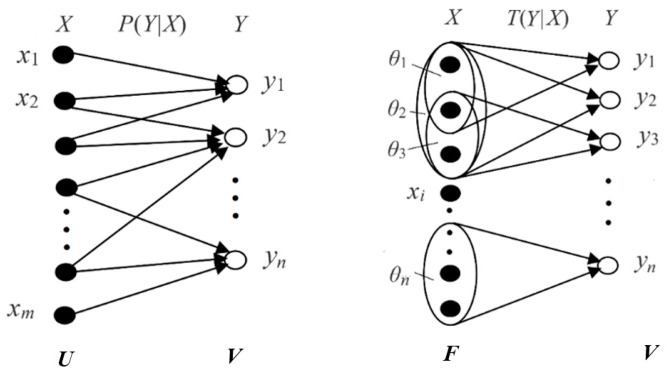
Shannon’s probability framework and the P-T probability framework.

**Figure 4 entropy-27-00461-f004:**
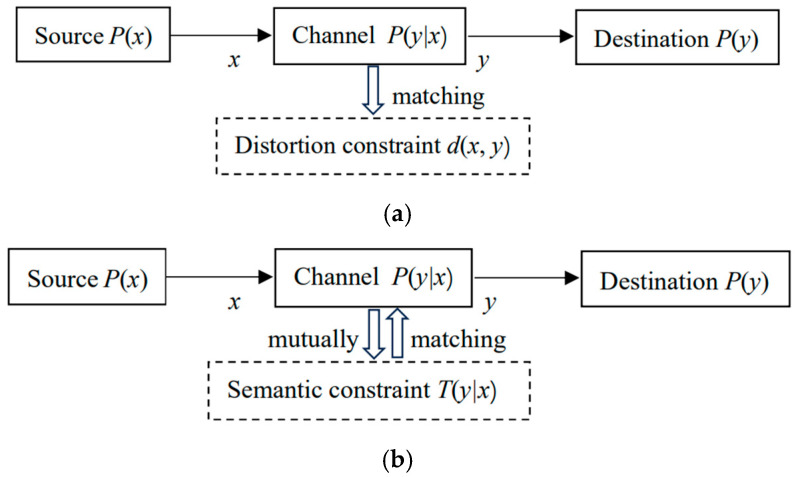
Communication models. (**a**) The Shannon communication model, where the channel needs to match the distortion function. (**b**) The semantic communication model, where two channels need to match mutually.

**Figure 5 entropy-27-00461-f005:**
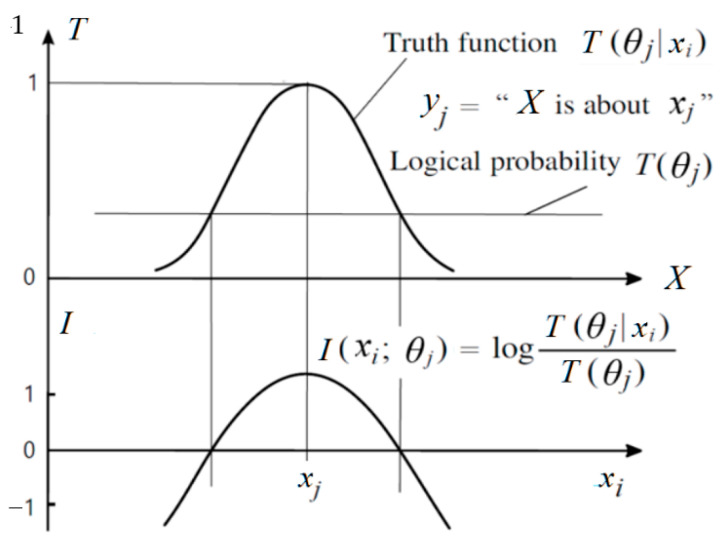
The semantic information *y_j_* conveys about *x_i_* decreases with the deviation increasing.

**Figure 6 entropy-27-00461-f006:**
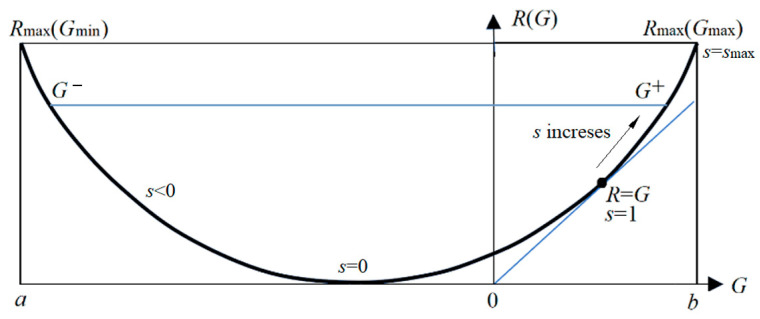
The information rate–fidelity function, *R*(*G*), for binary communication. Any *R*(*G*) function is a bowl-like function. There is a point at which *R*(*G*) = *G* (*s* = 1). Two anti-functions, *G*^−^(*R*) and *G*^+^(*R*), exist for a given *R*.

**Figure 7 entropy-27-00461-f007:**
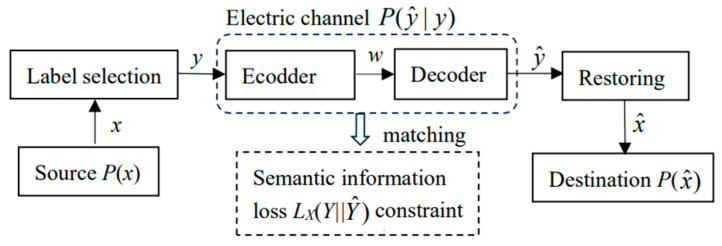
The electronic semantic communication model. The distortion constraint is replaced with the semantic information loss constraint. The $\hat{x}$ and $\hat{y}$ are estimates of *x* and *y*, and *w* is the electronic code to be transmitted.

**Figure 8 entropy-27-00461-f008:**
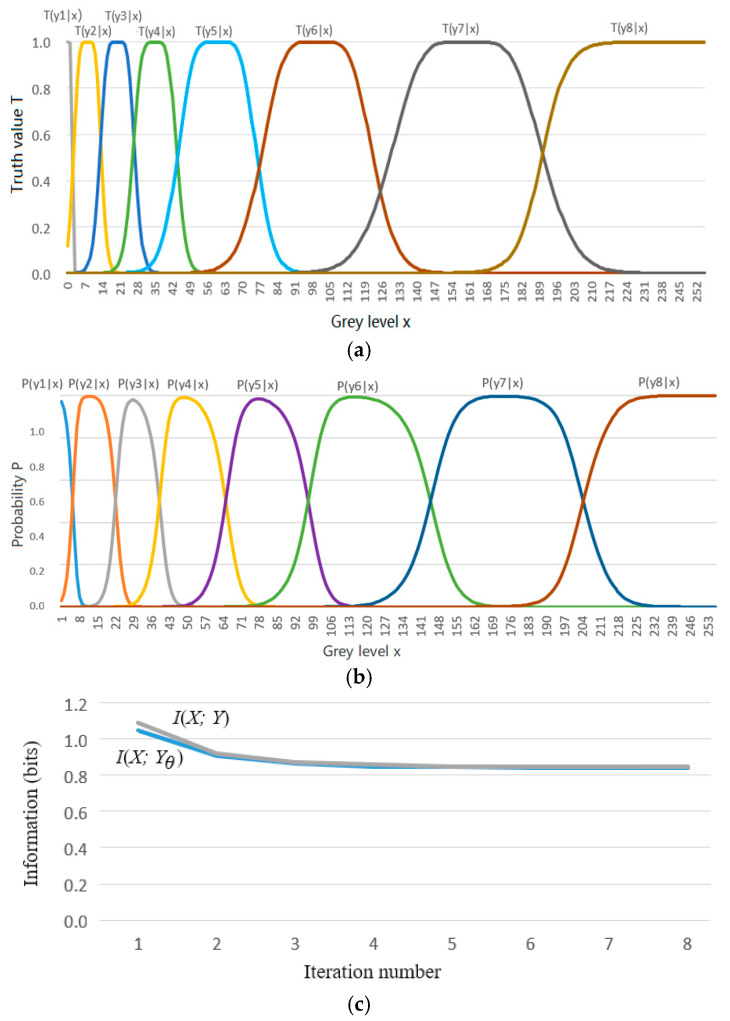
The gray level compression. (**a**) Eight truth functions form a semantic channel *T*(*y*|*x*) (see Appendix C in [37] for the data generation method). (**b**) Convergent Shannon channel *P*(*y*|*x*). (**c**) The variation of *I*(*X*; *Y_θ_*) and *I*(*X*; *Y*) during the iteration process.

**Figure 9 entropy-27-00461-f009:**
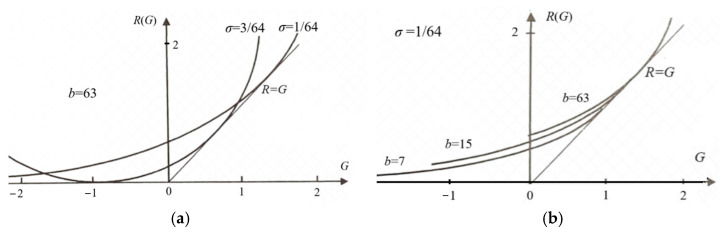
The variations of the *R*(*G*) function with discrimination (**a**) and quantization level (**b**).

**Figure 10 entropy-27-00461-f010:**
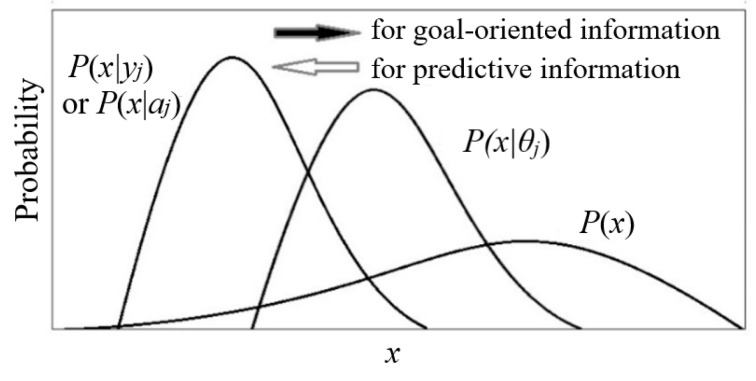
The optimization methods of the two types of semantic information are different. For predictive information, we hope that *P*(*x*|*θ_j_*) is close to *P*(*x*|*y_j_*) (see the white arrow), while for goal-oriented information, we hope that *P*(*x*|*a_j_*) is close to *P*(*x*|*θ_j_*) (see the black arrow).

**Figure 11 entropy-27-00461-f011:**
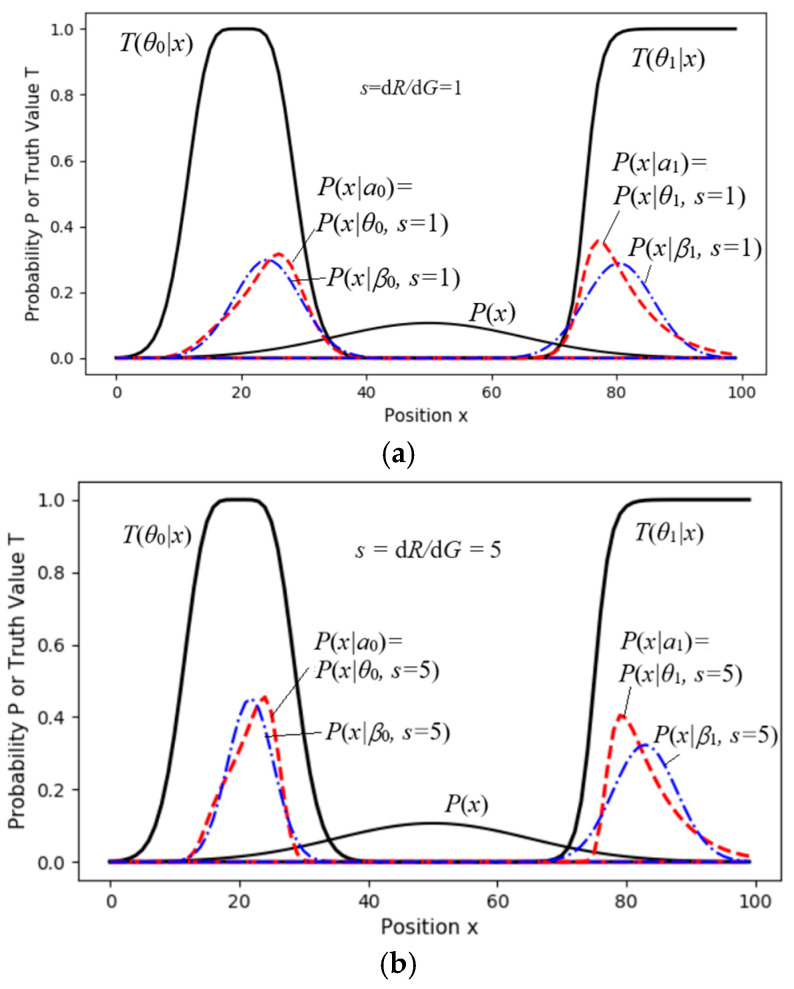
A two-objective control task. Dashed lines show *P*(*x*|*a_j_*) = *P*(*x*|*θ_j_*, *s*) (*j* = 0, 1), and dash–dotted lines represent *P*(*x*|*β_j_*, *s*) (*j* = 0, 1). (**a**) The case with *s* = 1; (**b**) The case with *s* = 5. *P*(*x*|*β_j_*, *s*) is a normal distribution produced by action *a_j_* (See [61] for details).

**Figure 12 entropy-27-00461-f012:**
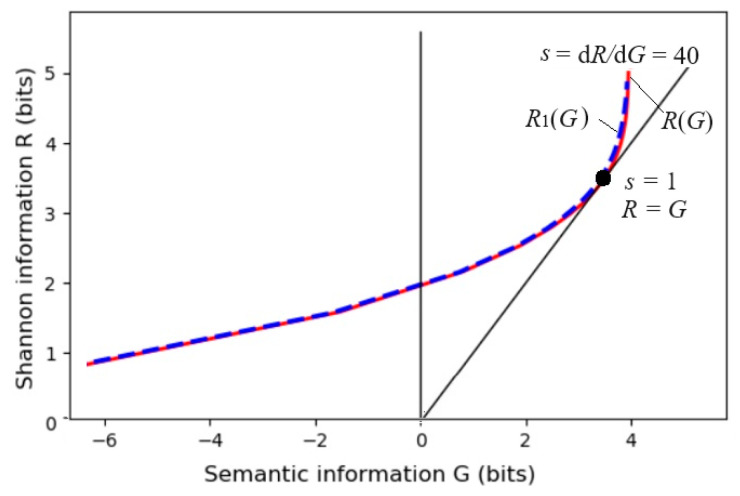
The *R*(*G*) for constraint control. *G* slightly increases when s increases from 5 to 40, meaning *s* = 5 is good enough.

**Figure 13 entropy-27-00461-f013:**
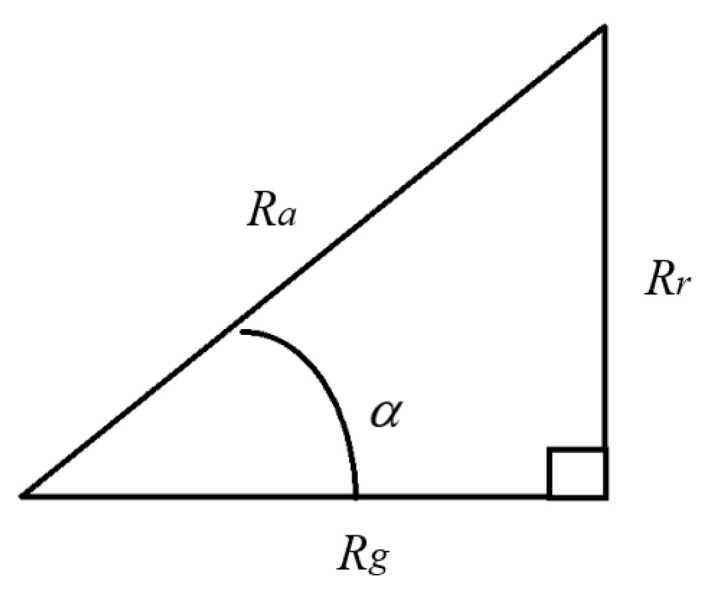
Relationship between the relative risk, sinα, and *R_r_*, *R_a_*, and *R_g_*.

**Figure 14 entropy-27-00461-f014:**
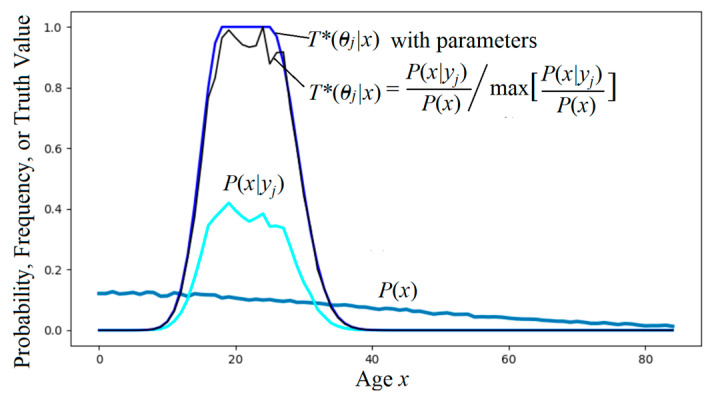
Using prior and posterior distributions, *P*(*x*) and *P*(*x*|*y_j_*), to obtain the optimized truth function, *T**(*θ_j_*|*x*). For details, see Appendix B in [7].

**Figure 15 entropy-27-00461-f015:**
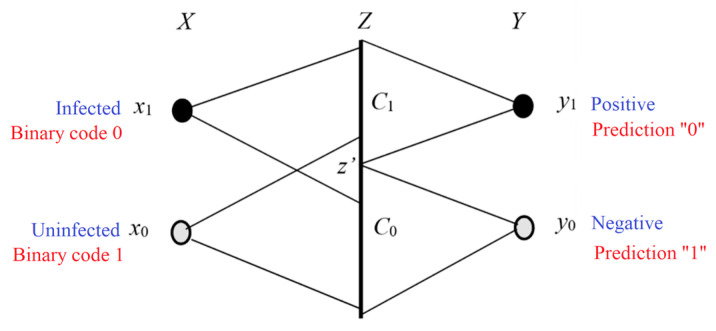
The medical test and the signal detection. We choose *y_j_* according to *z* ∈ *C_j_*. The task is to find the dividing point, *z*’, that results in maximum MI between *X* and *Y*.

**Figure 16 entropy-27-00461-f016:**
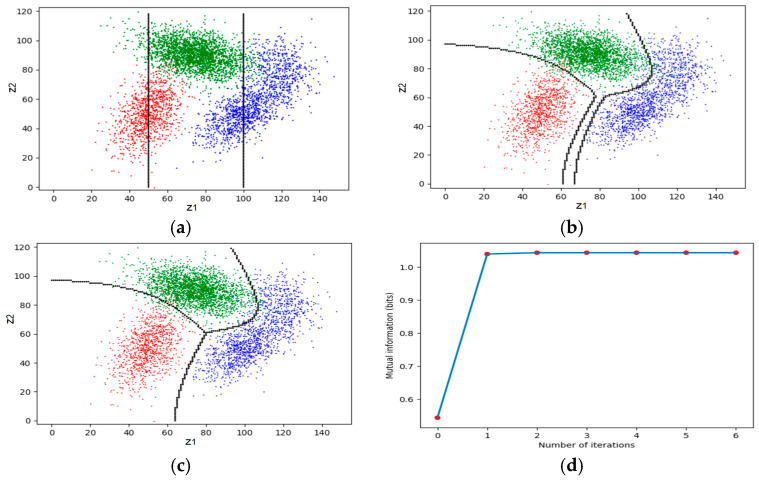
The maximum MI classification. Different colors represent different classes. (**a**) A very bad initial partition; (**b**) after the first iteration; (**c**) after the second iteration; (**d**) the MI changes with the iteration number.

**Figure 17 entropy-27-00461-f017:**
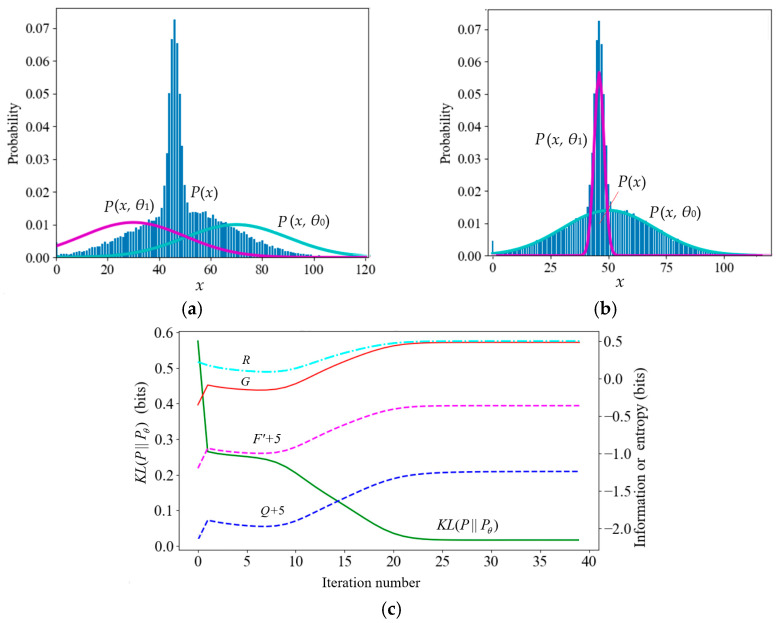
The convergent process of the mixture model from Neal and Hinton [21]. The mixture proportion was changed from 0.7:0.3 to 0.3:0.7. (**a**) The iteration starts; (**b**) the iteration converges; (**c**) the iteration process. *P*(*x*, *θ_j_*) equals *P*(*y_j_*)*P*(*x*|*θ_j_*) (*j* = 0, 1). See Appendix A for the Python source program.

**Figure 18 entropy-27-00461-f018:**
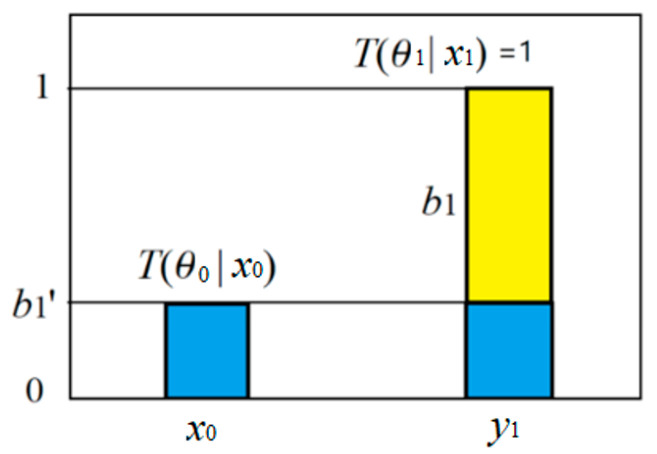
A truth function includes a believable proportion, *b*_1_′, and unbelievable proportion, *b*_1_′ = 1 − |*b*_1_|.

## Data Availability

Not applicable.

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
