# Peer review of "A Semantic Generalization of Shannon’s Information Theory and Applications"

_entropy, 2025, doi:10.3390/e27050461_

Round 1
Reviewer 1 Report (Previous Reviewer 2)
Comments and Suggestions for Authors
This paper can be accepted as it is.
Reviewer 2 Report (Previous Reviewer 1)
Comments and Suggestions for Authors
This paper is a resubmitted version of the manuscript that I have
already reviewed. The changes made are mostly cosmetic. Given the
time constraints, I found it impossible to traverse multiple threads
pursued in this paper. My general impression about the developed
theory of semantic information, called the G theory, is that it is
just a little and a bit arbitrary modification of concepts
discussed in the Shannon information theory. Given the breadth of the
perspective of the author, however, I do not mind a presentation of
this framework in Entropy as a form of documentation.
There are still some typos, such as "G theoryIn" in line 232 or
"P(X=\in\theta_j)" in line 259.
This manuscript is a resubmission of an earlier submission. The following is a list of the peer review reports and author responses from that submission.
Round 1
Reviewer 1 Report
Comments and Suggestions for Authors
The paper reviews a flavor of semantic information theory developed by
the author in a few previous publications. It is a comprehensive
40-pages' long survey. This paper is written from an engineer's
perspective, whereas I am a mathematician. Thus it speaks a bit
different language than I am accustomed to. It seems that semantic
information theory can have multiple engineering applications, in
machine learning in particular. However, from my point of view, it is
just a sort of a little restatement of Shannon's information theory
that can be obtained by regularizing information measures with a
certain KL divergence as in formula (15). It is kind of intriguing
that such a regularization can have an important impact onto
applications.
I must admit that the presentation was in crucial parts difficult to
understand for me. In particular, lines 200-213 should be written anew
gradually introducing Kolmogorov's discrete probability framework,
fuzzy sets and notation `"x\in\theta_j"` explicitly. The mapping
between y_j and theta_j should be also made more clear and
intuitive. At least, I got lost in the notations and I had to infer
the actual meaning of semantic information from later formulas and
examples, which also had a tendency of making simple things appear
more difficult than needed.
I agree that connections between semantic information theory and
algorithmic information theory (Kolmogorov complexity) are worth
exploring in the future (Section 6.4). I do not feel qualified to
assess the soundness of a wide range of other applications that the
author develops in Sections 3-6. However, I suppose that they may be
interesting to the audience of Entropy. Given the time constraints of
this review, I was not able to digest these applications enough to
formulate meaningful suggestions of amendments. The list and quality
of classical references that the author cites is impressive, ranging
from philosophy to engineering. I hope that these citations are meaningful
and on purpose. The discussion of the algorithmic information theory
in Section 6.4 could have been more thorough and precise, however.
Noticed typos:
In general, displayed formulas are of a varying height and a random
horizontal alignment, which is annoying while reading. I would highly
recommend that the author uses Latex rather than Word in future
publications. This yields neater typesetting.
Line 333: Bring -> Bringing
In the title of Section 6.3: Demantic -> Semantic
Reviewer 2 Report
Comments and Suggestions for Authors
This paper reviews the previous works in Shannon information, machine learning, and semantic communication. The author tries to link all these different theory into a single framework, which is called G theory. However, all the results are known and the authors failed to provide any new insights. There are already a large number of survey papers about semantic communication, most of which provide much more comprehensive coverage on semantic communication, machine learning, and information theory. There are even some books about semantic communication that have already been published. Therefore, the paper cannot be accepted in its current form. Some detailed comments are listed below:
(1) In the abstract and title, the author makes a bold statement claiming that this paper introduces the s-called “semantic information G theory”. First, it is very strange to use the term “G theory” for “generalization theory”. Second, information theory itself is a very general theory. In other words, semantic information theory already means a general theory for communications of semantic information and there is no need to introduce “semantic information generalization theory” which is incorrect both grammatically and semantically.
(2) The novelty of the paper is limited. All the theories and results have already been published a long time ago. Many recently published survey papers and books provide much more comprehensive coverages about semantic communication, compared to this paper.
(3) The mathematical definitions introduced in this paper are not consistent and do not link to each other. The author simply copies some math definitions and contents listed in some old papers without providing any new insights.
(4) The discussions provided in this paper lack depth in both theory and practice to support its claims.
Comments on the Quality of English LanguageSee previous comments.
Round 2
Reviewer 1 Report
Comments and Suggestions for Authors
The author took a large effort to improve the quality of the manuscript. I still do not fully grasp the practical utility of the exposed semantic information theory. However, I think that this extensive survey paper can be published in Entropy for future reference.
Reviewer 2 Report
Comments and Suggestions for Authors
The authors have done some revisions on the original manuscript. However, most of the major issues raised in my previous comments have not been addressed. It is quite obvious that the author has lost track of the recent 10-20 years of development in the semantic communication area, especially the task-oriented semantic communication. Also, the revised paper is still dominated by some very old concepts and theory. In addition to that, the paper is still lacking depth and coherent organization of many old materials. Detailed comments are given below:
- Most concepts reviewed in this paper are introduced at least 20 years ago. For example, the theory proposed by Carnap and Bar-Hillel was introduced at 1954 and even the newest content, the Florida's Strong Semantic Information Theory reviewed at the end of this paper is introduced in 2004, over 20 years ago. The progress in the past 20 years have been missed in this paper.
- Again, the author claims it introduce the so-called “semantic information G theory” which is a grammarly incorrect term. Information theory is itself is a very general theory. In the response, the author claims the letter "G" means "Generalizability", so the term “semantic information G theory” becomes "semantic information generalizability theory". This is obviously incorrect and has no meaning as the subject of the whole term falls into “information generalizability theory”.
The title of this paper contains grammarly incorrect term “semantic information G theory”. In the response, the author claims the letter "G" means "Generalizability", so the term “semantic information G theory” becomes "semantic information generalizability theory". This is incorrect and has no meaning as the subject of the whole term falls into “information generalizability theory”.
